# Targeted Attack against Deep Neural Networks via Flipping Limited Weight Bits

**Jiawang Bai** [1,2] [†]**, Baoyuan Wu** [3,4]**, Yong Zhang** [5]**, Yiming Li** [1]**, Zhifeng Li** [5]**, Shu-Tao Xia** [1,2]

[1] Tsinghua Shenzhen International Graduate School, Tsinghua University
[2] PCL Research Center of Networks and Communications, Peng Cheng Laboratory
[3] School of Data Science, The Chinese University of Hong Kong, Shenzhen
[4] Secure Computing Lab of Big Data, Shenzhen Research Institute of Big Data
[5] Tencent AI Lab

## Abstract

To explore the vulnerability of deep neural networks (DNNs), many attack paradigms have been well studied, such as the poisoning-based backdoor attack in the training stage and the adversarial attack in the inference stage. In this paper, we study a novel attack paradigm, which modifies model parameters in the deployment stage for malicious purposes. Specifically, our goal is to misclassify a specific sample into a target class without any sample modification, while not significantly reduce the prediction accuracy of other samples to ensure the stealthiness. To this end, we formulate this problem as a binary integer programming (BIP), since the parameters are stored as binary bits (*i.e.*, 0 and 1) in the memory. By utilizing the latest technique in integer programming, we equivalently reformulate this BIP problem as a continuous optimization problem, which can be effectively and efficiently solved using the alternating direction method of multipliers (ADMM) method. Consequently, the flipped critical bits can be easily determined through optimization, rather than using a heuristic strategy. Extensive experiments demonstrate the superiority of our method in attacking DNNs. The code is available at: `https://github.com/jiawangbai/TA-LBF`.

## 1 Introduction

Due to the great success of deep neural networks (DNNs), its vulnerability (Szegedy et al., 2014; Gu et al., 2019) has attracted great attention, especially for security-critical applications (*e.g.*, face recognition (Dong et al., 2019) and autonomous driving (Eykholt et al., 2018)). For example, backdoor attack (Saha et al., 2020; Xie et al., 2019) manipulates the behavior of the DNN model by mainly poisoning some training data in the training stage; adversarial attack (Goodfellow et al., 2015; Moosavi-Dezfooli et al., 2017) aims to fool the DNN model by adding malicious perturbations onto the input in the inference stage.

Compared to the backdoor attack and adversarial attack, a novel attack paradigm, dubbed *weight attack* (Breier et al., 2018), has been rarely studied. It assumes that the attacker has full access to the memory of a device, such that he/she can directly change the parameters of a deployed model to achieve some malicious purposes (*e.g.*, crushing a fully functional DNN and converting it to a random output generator (Rakin et al., 2019)). Since weight attack neither modifies the input nor control the training process, both the service provider and the user are difficult to realize the existence of the attack. In practice, since the deployed DNN model is stored as binary bits in the memory, the attacker can modify the model parameters using some physical fault injection techniques, such as Row Hammer Attack (Agoyan et al., 2010; Selmke et al., 2015) and Laser Beam Attack (Kim et al., 2014). These techniques can precisely flip any bit of the data in the memory. Some previous works (Rakin et al., 2019; 2020a;b) have demonstrated that it is feasible to change the model weights via bit flipping to achieve some malicious purposes. However, the critical bits are identified mostly

---

[†]This work was done when Jiawang Bai was an intern at Tencent AI Lab.
Correspondence to: Baoyuan Wu (wubaoyuan@cuhk.edu.cn) and Shu-Tao Xia (xiast@sz.tsinghua.edu.cn).

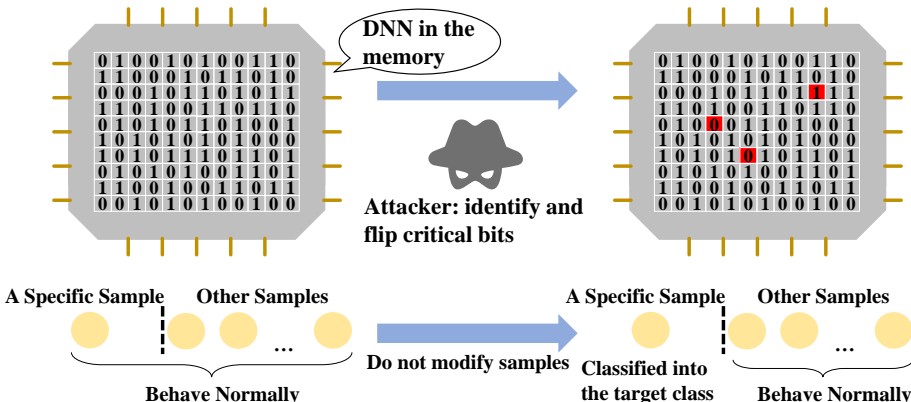

Figure 1: Demonstration of our proposed attack against a deployed DNN in the memory. By flipping critical bits (marked in red), our method can mislead a specific sample into the target class without any sample modification while not significantly reduce the prediction accuracy of other samples.

using some heuristic strategies in their methods. For example, Rakin et al. (2019) combined gradient ranking and progressive search to identify the critical bits for flipping.

This work also focuses on the bit-level weight attack against DNNs in the deployment stage, whereas with two different goals, including *effectiveness* and *stealthiness*. The effectiveness requires that the attacked model can misclassify a specific sample to a attacker-specified target class without any sample modification, while the stealthiness encourages that the prediction accuracy of other samples will not be significantly reduced. As shown in Fig. 1, to achieve these goals, we propose to identify and flip bits that are critical to the prediction of the specific sample but not significantly impact the prediction of other samples. Specifically, we treat each bit in the memory as a binary variable, and our task is to determine its state (*i.e.*, 0 or 1). Accordingly, it can be formulated as a binary integer programming (BIP) problem. To further improve the stealthiness, we also limit the number of flipped bits, which can be formulated as a cardinality constraint. However, how to solve the BIP problem with a cardinality constraint is a challenging problem. Fortunately, inspired by an advanced optimization method, the $\ell_p$-box ADMM (Wu & Ghanem, 2018), this problem can be reformulated as a continuous optimization problem, which can further be efficiently and effectively solved by the alternating direction method of multipliers (ADMM) (Glowinski & Marroco, 1975; Gabay & Mercier, 1976). Consequently, the flipped bits can be determined through optimization rather than the original heuristic strategy, which makes our attack more effective. Note that we also conduct attack against the quantized DNN models, following the setting in some related works (Rakin et al., 2019; 2020a). Extensive experiments demonstrate the superiority of the proposed method over several existing weight attacks. For example, our method achieves a 100% attack success rate with 7.37 bit-flips and 0.09% accuracy degradation of the rest unspecific inputs in attacking a 8-bit quantized ResNet-18 model on ImageNet. Moreover, we also demonstrate that the proposed method is also more resistant to existing defense methods.

The main contributions of this work are three-fold. **1)** We explore a novel attack scenario where the attacker enforces a specific sample to be predicted as a target class by modifying the weights of a deployed model via bit flipping without any sample modification. **2)** We formulate the attack as a BIP problem with the cardinality constraint and propose an effective and efficient method to solve this problem. **3)** Extensive experiments verify the superiority of the proposed method against DNNs with or without defenses.

## 2 RELATED WORKS

**Neural Network Weight Attack.** How to perturb the weights of a trained DNN for malicious purposes received extensive attention (Liu et al., 2017a; 2018b; Hong et al., 2019). Liu et al. (2017a) firstly proposed two schemes to modify model parameters for misclassification without and with considering stealthiness, which is dubbed single bias attack (SBA) and gradient descent

attack (GDA) respectively. After that, Trojan attack (Liu et al., 2018b) was proposed, which injects malicious behavior to the DNN by generating a general trojan trigger and then retraining the model. This method requires to change lots of parameters. Recently, fault sneaking attack (FSA) (Zhao et al., 2019) was proposed, which aims to misclassify certain samples into a target class by modifying the DNN parameters with two constraints, including maintaining the classification accuracy of other samples and minimizing parameter modifications. Note that all those methods are designed to misclassify multiple samples instead of a specific sample, which may probably modify lots of parameters or degrade the accuracy of other samples sharply.

**Bit-Flip based Attack.** Recently, some physical fault injection techniques (Agoyan et al., 2010; Kim et al., 2014; Selmke et al., 2015) were proposed, which can be adopted to precisely flip any bit in the memory. Those techniques promote researchers to study how to modify model parameters at the bit-level. As a branch of weight attack, the bit-flip based attack was firstly explored in (Rakin et al., 2019). It proposed an untargeted attack that can convert the attacked DNN to a random output generator with several bit-flips. Besides, Rakin et al. (2020a) proposed the targeted bit Trojan (TBT) to inject the fault into DNNs by flipping some critical bits. Specifically, the attacker flips the identified bits to force the network to classify all samples embedded with a trigger to a certain target class, while the network operates with normal inference accuracy with benign samples. Most recently, Rakin et al. (2020b) proposed the targeted bit-flip attack (T-BFA), which achieves malicious purposes without modifying samples. Specifically, T-BFA can mislead samples from single source class or all classes to a target class by flipping the identified weight bits. It is worth noting that the above bit-flip based attacks leverage heuristic strategies to identify critical weight bits. How to find critical bits for the bit-flip based attack method is still an important open question.

## 3 TARGETED ATTACK WITH LIMITED BIT-FLIPS (TA-LBF)

### 3.1 PRELIMINARIES

**Storage and Calculation of Quantized DNNs.** Currently, it is a widely-used technique to quantize DNNs before deploying on devices for efficiency and reducing storage size. For each weight in $l$-th layer of a Q-bit quantized DNN, it will be represented and then stored as the signed integer in two's complement representation ($\boldsymbol{v} = [v_Q; v_{Q-1}; ...; v_1] \in \{0, 1\}^Q$) in the memory. Attacker can modify the weights of DNNs through flipping the stored binary bits. In this work, we adopt the layer-wise uniform weight quantization scheme similar to Tensor-RT (Migacz, 2017). Accordingly, each binary vector $\boldsymbol{v}$ can be converted to a real number by a function $h(\cdot)$, as follow:

$$h(\boldsymbol{v}) = (-2^{Q-1} \cdot v_Q + \sum_{i=1}^{Q-1} 2^{i-1} \cdot v_i) \cdot \Delta^l, \tag{1}$$

where $l$ indicates which layer the weight is from, $\Delta^l > 0$ is a known and stored constant which represents the step size of the $l$-th layer weight quantizer.

**Notations.** We denote a Q-bit quantized DNN-based classification model as $f : \mathcal{X} \rightarrow \mathcal{Y}$, where $\mathcal{X} \in \mathbb{R}^d$ being the input space and $\mathcal{Y} \in \{1, 2, ..., K\}$ being the $K$-class output space. Assuming that the last layer of this DNN model is a fully-connected layer with $\mathbf{B} \in \{0, 1\}^{K \times C \times Q}$ being the quantized weights, where $C$ is the dimension of last layer's input. Let $\mathbf{B}_{i,j} \in \{0, 1\}^Q$ be the two's complement representation of a single weight and $\mathbf{B}_i \in \{0, 1\}^{C \times Q}$ denotes all the binary weights connected to the $i$-th output neuron. Given a test sample $\boldsymbol{x}$ with the ground-truth label $s$, $f(\boldsymbol{x}; \boldsymbol{\Theta}, \mathbf{B}) \in [0, 1]^K$ is the output probability vector and $g(\boldsymbol{x}; \boldsymbol{\Theta}) \in \mathbb{R}^C$ is the input of the last layer, where $\boldsymbol{\Theta}$ denotes the model parameters without the last layer.

**Attack Scenario.** In this paper, we focus on the white-box bit-flip based attack, which was first introduced in (Rakin et al., 2019). Specifically, we assume that the attacker has full knowledge of the model (including it's architecture, parameters, and parameters' location in the memory), and can precisely flip any bit in the memory. Besides, we also assume that attackers can have access to a small portion of benign samples, but they can not tamper the training process and the training data.

**Attacker's Goals.** Attackers have two main goals, including the *effectiveness* and the *stealthiness*. Specifically, *effectiveness* requires that the attacked model can misclassify a specific sample to a predefined target class without any sample modification, and the *stealthiness* requires that the prediction accuracy of other samples will not be significantly reduced.

## 3.2 THE PROPOSED METHOD

**Loss for Ensuring Effectiveness.** Recall that our first target is to force a specific image to be classified as the target class by modifying the model parameters at the bit-level. To this end, the most straightforward way is maximizing the logit of the target class while minimizing that of the source class. For a sample $\boldsymbol{x}$, the logit of a class can be directly determined by the input of the last layer $g(\boldsymbol{x}; \boldsymbol{\Theta})$ and weights connected to the node of that class. Accordingly, we can modify weights only connected to the source and target class to fulfill our purpose, as follows:

$$\mathcal{L}_1(\boldsymbol{x}; \boldsymbol{\Theta}, \mathbf{B}, \hat{\mathbf{B}}_s, \hat{\mathbf{B}}_t) = \max\left(m - p(\boldsymbol{x}; \boldsymbol{\Theta}, \hat{\mathbf{B}}_t) + \delta, 0\right) + \max\left(p(\boldsymbol{x}; \boldsymbol{\Theta}, \hat{\mathbf{B}}_s) - m + \delta, 0\right), \quad (2)$$

where $p(\boldsymbol{x}; \boldsymbol{\Theta}, \hat{\mathbf{B}}_i) = [h(\hat{\mathbf{B}}_{i,1}); h(\hat{\mathbf{B}}_{i,2}); ...; h(\hat{\mathbf{B}}_{i,C})]^\top g(\boldsymbol{x}; \boldsymbol{\Theta})$ denotes the logit of class $i$ ($i = s$ or $i = t$), $h(\cdot)$ is the function defined in Eq. (1), $m = \max\limits_{i \in \{0,...,K\} \backslash \{s\}} p(\boldsymbol{x}; \boldsymbol{\Theta}, \mathbf{B}_i)$, and $\delta \in \mathbb{R}$ indicates a slack variable, which will be specified in later experiments. The first term of $\mathcal{L}_1$ aims at increasing the logit of the target class, while the second term is to decrease the logit of the source class. The loss $\mathcal{L}_1$ is 0 only when the output on target class is more than $m + \delta$ and the output on source class is less than $m - \delta$. That is, the prediction on $\boldsymbol{x}$ of the target model is the predefined target class. Note that $\hat{\mathbf{B}}_s, \hat{\mathbf{B}}_t \in \{0,1\}^{C \times Q}$ are two variables we want to optimize, corresponding to the weights of the fully-connected layer $w.r.t.$ class $s$ and $t$, respectively, in the target DNN model. $\mathbf{B} \in \{0,1\}^{K \times C \times Q}$ denotes the weights of the fully-connected layer of the original DNN model, and it is a constant tensor in $\mathcal{L}_1$. For clarity, hereafter we simplify $\mathcal{L}_1(\boldsymbol{x}; \boldsymbol{\Theta}, \mathbf{B}, \hat{\mathbf{B}}_s, \hat{\mathbf{B}}_t)$ as $\mathcal{L}_1(\hat{\mathbf{B}}_s, \hat{\mathbf{B}}_t)$, since $\boldsymbol{x}$ and $\boldsymbol{\Theta}$ are also provided input and weights.

**Loss for Ensuring Stealthiness.** As we mentioned in Section 3.1, we assume that the attacker can get access to an auxiliary sample set $\{(\boldsymbol{x}_i, y_i)\}_{i=1}^N$. Accordingly, the stealthiness of the attack can be formulated as follows:

$$\mathcal{L}_2(\hat{\mathbf{B}}_s, \hat{\mathbf{B}}_t) = \sum_{i=1}^N \ell(f(\boldsymbol{x}_i; \boldsymbol{\Theta}, \mathbf{B}_{\{1,...,K\} \backslash \{s,t\}}, \hat{\mathbf{B}}_s, \hat{\mathbf{B}}_t), y_i), \quad (3)$$

where $\mathbf{B}_{\{1,...,K\} \backslash \{s,t\}}$ denotes $\{\mathbf{B}_1, \mathbf{B}_2, ..., \mathbf{B}_K\} \backslash \{\mathbf{B}_s, \mathbf{B}_t\}$, and $f_j(\boldsymbol{x}_i; \boldsymbol{\Theta}, \mathbf{B}_{\{1,...,K\} \backslash \{s,t\}}, \hat{\mathbf{B}}_s, \hat{\mathbf{B}}_t)$ indicates the posterior probability of $\boldsymbol{x}_i$ $w.r.t.$ class $j$, caclulated by $\text{Softmax}(p(\boldsymbol{x}_i; \boldsymbol{\Theta}, \hat{\mathbf{B}}_j))$ or $\text{Softmax}(p(\boldsymbol{x}_i; \boldsymbol{\Theta}, \mathbf{B}_j))$. $\ell(\cdot, \cdot)$ is specified by the cross entropy loss. To keep clarity, $\boldsymbol{x}_i$, $\boldsymbol{\Theta}$ and $\mathbf{B}_{\{1,...,K\} \backslash \{s,t\}}$ are omitted in $\mathcal{L}_2(\hat{\mathbf{B}}_s, \hat{\mathbf{B}}_t)$ .

Besides, to better meet our goal, a straightforward additional approach is reducing the magnitude of the modification. In this paper, we constrain the number of bit-flips less than $k$. Physical bit flipping techniques can be time-consuming as discussed in (Van Der Veen et al., 2016; Zhao et al., 2019). Moreover, such techniques lead to abnormal behaviors in the attacked system (*e.g.*, suspicious cache activity of processes), which may be detected by some physical detection-based defenses (Gruss et al., 2018). As such, minimizing the number of bit-flips is critical to make the attack more efficient and practical.

**Overall Objective.** In conclusion, the final objective function is as follows:

$$\min_{\hat{\mathbf{B}}_s, \hat{\mathbf{B}}_t} \mathcal{L}_1(\hat{\mathbf{B}}_s, \hat{\mathbf{B}}_t) + \lambda \mathcal{L}_2(\hat{\mathbf{B}}_s, \hat{\mathbf{B}}_t),$$
$$\text{s.t. } \hat{\mathbf{B}}_s \in \{0,1\}^{C \times Q}, \ \hat{\mathbf{B}}_t \in \{0,1\}^{C \times Q}, \ d_H(\mathbf{B}_s, \hat{\mathbf{B}}_s) + d_H(\mathbf{B}_t, \hat{\mathbf{B}}_t) \le k, \quad (4)$$

where $d_H(\cdot, \cdot)$ denotes the Hamming distance and $\lambda > 0$ is a trade-off parameter.

For the sake of brevity, $\mathbf{B}_s$ and $\mathbf{B}_t$ are concatenated and further reshaped to the vector $\boldsymbol{b} \in \{0,1\}^{2CQ}$. Similarly, $\hat{\mathbf{B}}_s$ and $\hat{\mathbf{B}}_t$ are concatenated and further reshaped to the vector $\hat{\boldsymbol{b}} \in \{0,1\}^{2CQ}$. Besides, for binary vector $\boldsymbol{b}$ and $\hat{\boldsymbol{b}}$, there exists a nice relationship between Hamming distance and Euclidean distance: $d_H(\boldsymbol{b}, \hat{\boldsymbol{b}}) = ||\boldsymbol{b} - \hat{\boldsymbol{b}}||_2^2$. The new formulation of the objective is as follows:

$$\min_{\hat{\boldsymbol{b}}} \mathcal{L}_1(\hat{\boldsymbol{b}}) + \lambda \mathcal{L}_2(\hat{\boldsymbol{b}}), \quad \text{s.t. } \hat{\boldsymbol{b}} \in \{0,1\}^{2CQ}, \ ||\boldsymbol{b} - \hat{\boldsymbol{b}}||_2^2 - k \le 0. \quad (5)$$

Problem (5) is denoted as TA-LBF (targeted attack with limited bit-flips). Note that TA-LBF is a binary integer programming (BIP) problem, whose optimization is challenging. We will introduce an effective and efficient method to solve it in the following section.

### 3.3 AN EFFECTIVE OPTIMIZATION METHOD FOR TA-LBF

To solve the challenging BIP problem (5), we adopt the generic solver for integer programming, dubbed $\ell_p$-Box ADMM (Wu & Ghanem, 2018). The solver presents its superior performance in many tasks, $e.g.$, model pruning (Li et al., 2019), clustering (Bibi et al., 2019), MAP inference (Wu et al., 2020a), adversarial attack (Fan et al., 2020), $etc.$. It proposed to replace the binary constraint equivalently by the intersection of two continuous constraints, as follows

$$\hat{\boldsymbol{b}} \in \{0,1\}^{2CQ} \Leftrightarrow \hat{\boldsymbol{b}} \in (\mathcal{S}_b \cap \mathcal{S}_p), \tag{6}$$

where $\mathcal{S}_b = [0,1]^{2CQ}$ indicates the box constraint, and $\mathcal{S}_p = \{\hat{\boldsymbol{b}} : ||\hat{\boldsymbol{b}} - \frac{1}{2}||_2^2 = \frac{2CQ}{4}\}$ denotes the $\ell_2$-sphere constraint. Utilizing (6), Problem (5) is equivalently reformulated as

$$\min_{\hat{\boldsymbol{b}}, \boldsymbol{u}_1 \in \mathcal{S}_b, \boldsymbol{u}_2 \in \mathcal{S}_p, u_3 \in \mathbb{R}^+} \mathcal{L}_1(\hat{\boldsymbol{b}}) + \lambda \mathcal{L}_2(\hat{\boldsymbol{b}}), \quad \text{s.t.} \ \hat{\boldsymbol{b}} = \boldsymbol{u}_1, \hat{\boldsymbol{b}} = \boldsymbol{u}_2, ||\boldsymbol{b} - \hat{\boldsymbol{b}}||_2^2 - k + u_3 = 0, \tag{7}$$

where two extra variables $\boldsymbol{u}_1$ and $\boldsymbol{u}_2$ are introduced to split the constraints $w.r.t.$ $\hat{\boldsymbol{b}}$. Besides, the non-negative slack variable $u_3 \in \mathbb{R}^+$ is used to transform $||\boldsymbol{b} - \hat{\boldsymbol{b}}||_2^2 - k \le 0$ in (5) into $||\boldsymbol{b} - \hat{\boldsymbol{b}}||_2^2 - k + u_3 = 0$. The above constrained optimization problem can be efficiently solved by the alternating direction method of multipliers (ADMM) (Boyd et al., 2011).

Following the standard procedure of ADMM, we firstly present the augmented Lagrangian function of the above problem, as follows:

$$
\begin{aligned}
L(\hat{\boldsymbol{b}}, \boldsymbol{u}_1, \boldsymbol{u}_2, u_3, \boldsymbol{z}_1, \boldsymbol{z}_2, z_3) =& \mathcal{L}_1(\hat{\boldsymbol{b}}) + \lambda \mathcal{L}_2(\hat{\boldsymbol{b}}) + \boldsymbol{z}_1^\top(\hat{\boldsymbol{b}} - \boldsymbol{u}_1) + \boldsymbol{z}_2^\top(\hat{\boldsymbol{b}} - \boldsymbol{u}_2) \\
&+ z_3(||\boldsymbol{b} - \hat{\boldsymbol{b}}||_2^2 - k + u_3) + c_1(\boldsymbol{u}_1) + c_2(\boldsymbol{u}_2) + c_3(u_3) \\
&+ \frac{\rho_1}{2}||\hat{\boldsymbol{b}} - \boldsymbol{u}_1||_2^2 + \frac{\rho_2}{2}||\hat{\boldsymbol{b}} - \boldsymbol{u}_2||_2^2 + \frac{\rho_3}{2}(||\boldsymbol{b} - \hat{\boldsymbol{b}}||_2^2 - k + u_3)^2,
\end{aligned}
\tag{8}
$$

where $\boldsymbol{z}_1, \boldsymbol{z}_2 \in \mathbb{R}^{2CQ}$ and $z_3 \in \mathbb{R}$ are dual variables, and $\rho_1, \rho_2, \rho_3 > 0$ are penalty factors, which will be specified later. $c_1(\boldsymbol{u}_1) = \mathbb{I}_{\{\boldsymbol{u}_1 \in \mathcal{S}_b\}}$, $c_2(\boldsymbol{u}_2) = \mathbb{I}_{\{\boldsymbol{u}_2 \in \mathcal{S}_p\}}$, and $c_3(u_3) = \mathbb{I}_{\{u_3 \in \mathbb{R}^+\}}$ capture the constraints $\mathcal{S}_b, \mathcal{S}_p$ and $\mathbb{R}^+$, respectively. The indicator function $\mathbb{I}_{\{a\}} = 0$ if $a$ is true; otherwise, $\mathbb{I}_{\{a\}} = +\infty$. Based on the augmented Lagrangian function, the primary and dual variables are updated iteratively, with $r$ indicating the iteration index.

**Given $(\hat{\boldsymbol{b}}^r, \boldsymbol{z}_1^r, \boldsymbol{z}_2^r, z_3^r)$, update $(\boldsymbol{u}_1^{r+1}, \boldsymbol{u}_2^{r+1}, u_3^{r+1})$.** Given $(\hat{\boldsymbol{b}}^r, \boldsymbol{z}_1^r, \boldsymbol{z}_2^r, z_3^r)$, $(\boldsymbol{u}_1, \boldsymbol{u}_2, u_3)$ are independent, and they can be optimized in parallel, as follows

$$
\begin{cases}
\boldsymbol{u}_1^{r+1} = \underset{\boldsymbol{u}_1 \in \mathcal{S}_b}{\arg\min} \ (\boldsymbol{z}_1^r)^\top(\hat{\boldsymbol{b}}^r - \boldsymbol{u}_1) + \frac{\rho_1}{2}||\hat{\boldsymbol{b}}^r - \boldsymbol{u}_1||_2^2 = \mathcal{P}_{\mathcal{S}_b}(\hat{\boldsymbol{b}}^r + \frac{\boldsymbol{z}_1^r}{\rho_1}), \\
\boldsymbol{u}_2^{r+1} = \underset{\boldsymbol{u}_2 \in \mathcal{S}_p}{\arg\min} \ (\boldsymbol{z}_2^r)^\top(\hat{\boldsymbol{b}}^r - \boldsymbol{u}_2) + \frac{\rho_2}{2}||\hat{\boldsymbol{b}}^r - \boldsymbol{u}_2||_2^2 = \mathcal{P}_{\mathcal{S}_p}(\hat{\boldsymbol{b}}^r + \frac{\boldsymbol{z}_2^r}{\rho_2}), \\
u_3^{r+1} = \underset{u_3 \in \mathbb{R}^+}{\arg\min} \ z_3^r(||\boldsymbol{b} - \hat{\boldsymbol{b}}^r||_2^2 - k + u_3) + \frac{\rho_3}{2}(||\boldsymbol{b} - \hat{\boldsymbol{b}}^r||_2^2 - k + u_3)^2 \\
\qquad = \mathcal{P}_{\mathbb{R}^+}(-||\boldsymbol{b} - \hat{\boldsymbol{b}}^r||_2^2 + k - \frac{z_3^r}{\rho_3}),
\end{cases}
\tag{9}
$$

where $\mathcal{P}_{\mathcal{S}_b}(\boldsymbol{a}) = \min((\boldsymbol{1}, \max(\boldsymbol{0}, \boldsymbol{a})))$ with $\boldsymbol{a} \in \mathbb{R}^n$ is the projection onto the box constraint $\mathcal{S}_b$; $\mathcal{P}_{\mathcal{S}_p}(\boldsymbol{a}) = \frac{\sqrt{n}}{2} \frac{\bar{\boldsymbol{a}}}{||\bar{\boldsymbol{a}}||} + \frac{1}{2}$ with $\bar{\boldsymbol{a}} = \boldsymbol{a} - \frac{1}{2}$ indicates the projection onto the $\ell_2$-sphere constraint $\mathcal{S}_p$ (Wu & Ghanem, 2018); $\mathcal{P}_{\mathbb{R}^+}(a) = \max(0, a)$ with $a \in \mathbb{R}$ indicates the projection onto $\mathbb{R}^+$.

**Given $(\boldsymbol{u}_1^{r+1}, \boldsymbol{u}_2^{r+1}, u_3^{r+1}, \boldsymbol{z}_1^r, \boldsymbol{z}_2^r, z_3^r)$, update $\hat{\boldsymbol{b}}^{r+1}$.** Although there is no closed-form solution to $\hat{\boldsymbol{b}}^{r+1}$, it can be easily updated by the gradient descent method, as both $\mathcal{L}_1(\hat{\boldsymbol{b}})$ and $\mathcal{L}_2(\hat{\boldsymbol{b}})$ are differentiable $w.r.t.$ $\hat{\boldsymbol{b}}$, as follows

$$\hat{\boldsymbol{b}}^{r+1} \leftarrow \hat{\boldsymbol{b}}^r - \eta \cdot \frac{\partial L(\hat{\boldsymbol{b}}, \boldsymbol{u}_1^{r+1}, \boldsymbol{u}_2^{r+1}, u_3^{r+1}, \boldsymbol{z}_1^r, \boldsymbol{z}_2^r, z_3^r)}{\partial \hat{\boldsymbol{b}}}\bigg|_{\hat{\boldsymbol{b}} = \hat{\boldsymbol{b}}^r}, \tag{10}$$

where $\eta > 0$ denotes the step size. Note that we can run multiple steps of gradient descent in the above update. Both the number of steps and $\eta$ will be specified in later experiments. Besides, due to the space limit, the detailed derivation of $\partial L / \partial \hat{\boldsymbol{b}}$ will be presented in **Appendix A**.

**Given** $(\hat{\boldsymbol{b}}^{r+1}, \boldsymbol{u}_1^{r+1}, \boldsymbol{u}_2^{r+1}, u_3^{r+1})$, **update** $(\boldsymbol{z}_1^{r+1}, \boldsymbol{z}_2^{r+1}, z_3^{r+1})$. The dual variables are updated by the gradient ascent method, as follows

$$\begin{cases} \boldsymbol{z}_1^{r+1} = \boldsymbol{z}_1^r + \rho_1(\hat{\boldsymbol{b}}^{r+1} - \boldsymbol{u}_1^{r+1}), \\ \boldsymbol{z}_2^{r+1} = \boldsymbol{z}_2^r + \rho_2(\hat{\boldsymbol{b}}^{r+1} - \boldsymbol{u}_2^{r+1}), \\ z_3^{r+1} = z_3^r + \rho_3(||\boldsymbol{b} - \hat{\boldsymbol{b}}^{r+1}||_2^2 - k + u_3^{r+1}). \end{cases} \quad (11)$$

**Remarks.** **1)** Note that since $(\boldsymbol{u}_1^{r+1}, \boldsymbol{u}_2^{r+1}, u_3^{r+1})$ are updated in parallel, their updates belong to the same block. Thus, the above algorithm is a two-block ADMM algorithm. We provide the algorithm outline in **Appendix B**. **2)** Except for the update of $\hat{\boldsymbol{b}}^{r+1}$, all other updates are very simple and efficient. The computational cost of the whole algorithm will be analyzed in **Appendix C**. **3)** Due to the inexact solution to $\hat{\boldsymbol{b}}^{r+1}$ using gradient descent, the theoretical convergence of the whole ADMM algorithm cannot be guaranteed. However, as demonstrated in many previous works (Gol'shtein & Tret'yakov, 1979; Eckstein & Bertsekas, 1992; Boyd et al., 2011), the inexact two-block ADMM often shows good practical convergence, which is also the case in our later experiments. Besides, the numerical convergence analysis is presented in **Appendix D**. **4)** The proper adjustment of $(\rho_1, \rho_2, \rho_3)$ could accelerate the practical convergence, which will be specified later .

## 4 Experiments

### 4.1 Evaluation Setup

**Settings.** We compare our method (TA-LBF) with GDA (Liu et al., 2017a), FSA (Zhao et al., 2019), T-BFA (Rakin et al., 2020b), and TBT (Rakin et al., 2020a). All those methods can be adopted to misclassify a specific image into a target class. We also take the fine-tuning (FT) of the last fully-connected layer as a baseline method. We conduct experiments on CIFAR-10 (Krizhevsky et al., 2009) and ImageNet (Russakovsky et al., 2015). We randomly select 1,000 images from each dataset as the evaluation set for all methods. Specifically, for each of the 10 classes in CIFAR-10, we perform attacks on the 100 randomly selected validation images from the other 9 classes. For ImageNet, we randomly choose 50 target classes. For each target class, we perform attacks on 20 images randomly selected from the rest classes in the validation set. Besides, for all methods except GDA which does not employ auxiliary samples, we provide 128 and 512 auxiliary samples on CIFAR-10 and ImageNet, respectively. Following the setting in (Rakin et al., 2020a;b), we adopt the quantized ResNet (He et al., 2016) and VGG (Simonyan & Zisserman, 2015) as the target models. For our TA-LBF, the trade-off parameter $\lambda$ and the constraint parameter $k$ affect the attack stealthiness and the attack success rate. We adopt a strategy for jointly searching $\lambda$ and $k$, which is specified in **Appendix E.3**. More descriptions of our settings are provided in **Appendix E**.

**Evaluation Metrics.** We adopt three metrics to evaluate the attack performance, *i.e.,* the post attack accuracy (PA-ACC), the attack success rate (ASR), and the number of bit-flips ($N_{flip}$). PA-ACC denotes the post attack accuracy on the validation set except for the specific attacked sample and the auxiliary samples. ASR is defined as the ratio of attacked samples that are successfully attacked into the target class among all 1,000 attacked samples. $N_{flip}$ is the number of bit-flips required for an attack. A better attack performance corresponds to a higher PA-ACC and ASR, while a lower $N_{flip}$. Besides, we also show the accuracy of the original model, denoted as ACC.

### 4.2 Main Results

**Results on CIFAR-10.** The results of all methods on CIFAR-10 are shown in Table 1. Our method achieves a 100% ASR with the fewest $N_{flip}$ for all the bit-widths and architectures. FT modifies the maximum number of bits among all methods since there is no limitation of parameter modifications. Due to the absence of the training data, the PA-ACC of FT is also poor. These results indicate that fine-tuning the trained DNN as an attack method is infeasible. Although T-BFA flips the second-fewest bits under three cases, it fails to achieve a higher ASR than GDA and FSA. In terms of PA-ACC, TA-LBF is comparable to other methods. Note that the PA-ACC of TA-LBF significantly outperforms that of GDA, which is the most competitive *w.r.t.* ASR and $N_{flip}$ among all the baseline methods. The PA-ACC of GDA is relatively poor, because it does not employ auxiliary samples. Achieving the highest ASR, the lowest $N_{flip}$, and the comparable PA-ACC demonstrates that our optimization-based method is more superior than other heuristic methods (TBT, T-BFA and GDA).

Table 1: Results of all attack methods across different bit-widths and architectures on CIFAR-10 and ImageNet (bold: the best; underline: the second best). The mean and standard deviation of PA-ACC and $N_{flip}$ are calculated by attacking the 1,000 images. Our method is denoted as **TA-LBF**.

| Dataset | Method | Target Model | PA-ACC (%) | ASR (%) | $N_{flip}$ | Target Model | PA-ACC (%) | ASR (%) | $N_{flip}$ |
|---|---|---|---|---|---|---|---|---|---|
| CIFAR-10 | FT | ResNet 8-bit | $85.01_{\pm2.90}$ | **100.0** | $1507.51_{\pm86.54}$ | VGG 8-bit | $84.31_{\pm3.10}$ | 98.7 | $11298.74_{\pm830.36}$ |
| | TBT | | $88.07_{\pm0.84}$ | 97.3 | $246.70_{\pm8.19}$ | | $77.79_{\pm23.35}$ | 51.6 | $599.40_{\pm19.53}$ |
| | T-BFA | | $87.56_{\pm2.22}$ | 98.7 | $\underline{9.91_{\pm2.33}}$ | | $\mathbf{89.83_{\pm3.92}}$ | 96.7 | $\underline{14.53_{\pm3.74}}$ |
| | FSA | ACC: 92.16% | $\mathbf{88.38_{\pm2.28}}$ | 98.9 | $185.51_{\pm54.93}$ | ACC: 93.20% | $88.80_{\pm2.86}$ | 96.8 | $253.92_{\pm122.06}$ |
| | GDA | | $86.73_{\pm3.50}$ | 99.8 | $26.83_{\pm12.50}$ | | $85.51_{\pm2.88}$ | **100.0** | $21.54_{\pm6.79}$ |
| | **TA-LBF** | | $88.20_{\pm2.64}$ | **100.0** | $\mathbf{5.57_{\pm1.58}}$ | | $86.06_{\pm3.17}$ | **100.0** | $\mathbf{7.40_{\pm2.72}}$ |
| | FT | ResNet 4-bit | $84.37_{\pm2.94}$ | **100.0** | $392.48_{\pm47.26}$ | VGG 4-bit | $83.31_{\pm3.76}$ | 94.5 | $2270.52_{\pm324.69}$ |
| | TBT | | $\underline{87.79_{\pm1.86}}$ | 96.0 | $118.20_{\pm15.03}$ | | $83.90_{\pm2.63}$ | 62.4 | $266.40_{\pm18.70}$ |
| | T-BFA | | $86.46_{\pm2.80}$ | 97.9 | $\underline{8.80_{\pm2.01}}$ | | $\mathbf{88.74_{\pm4.52}}$ | 96.2 | $11.23_{\pm2.36}$ |
| | FSA | ACC: 91.90% | $87.73_{\pm2.36}$ | 98.4 | $76.83_{\pm25.27}$ | ACC: 92.61% | $\underline{87.58_{\pm3.06}}$ | 97.5 | $75.03_{\pm29.75}$ |
| | GDA | | $86.25_{\pm3.59}$ | 99.8 | $14.08_{\pm7.94}$ | | $85.08_{\pm2.82}$ | **100.0** | $\underline{10.31_{\pm3.77}}$ |
| | **TA-LBF** | | $\mathbf{87.82_{\pm2.60}}$ | **100.0** | $\mathbf{5.25_{\pm1.09}}$ | | $85.91_{\pm3.29}$ | **100.0** | $\mathbf{6.26_{\pm2.37}}$ |
| ImageNet | FT | ResNet 8-bit | $59.33_{\pm0.93}$ | **100.0** | $277424.29_{\pm12136.34}$ | VGG 8-bit | $62.08_{\pm2.33}$ | **100.0** | $1729685.22_{\pm137539.54}$ |
| | TBT | | $69.18_{\pm0.03}$ | 99.9 | $577.40_{\pm19.42}$ | | $72.99_{\pm0.02}$ | 99.2 | $4115.26_{\pm191.25}$ |
| | T-BFA | | $68.71_{\pm0.36}$ | 79.3 | $24.57_{\pm20.03}$ | | $73.09_{\pm0.12}$ | 84.5 | $363.78_{\pm153.28}$ |
| | FSA | ACC: 69.50% | $\underline{69.27_{\pm0.15}}$ | 99.7 | $441.21_{\pm119.45}$ | ACC: 73.31% | $\underline{73.28_{\pm0.03}}$ | **100.0** | $1030.03_{\pm260.30}$ |
| | GDA | | $69.26_{\pm0.22}$ | **100.0** | $\underline{18.54_{\pm6.14}}$ | | $\mathbf{73.29_{\pm0.02}}$ | **100.0** | $\underline{197.05_{\pm49.85}}$ |
| | **TA-LBF** | | $\mathbf{69.41_{\pm0.08}}$ | **100.0** | $\mathbf{7.37_{\pm2.18}}$ | | $73.28_{\pm0.03}$ | **100.0** | $\mathbf{69.89_{\pm18.42}}$ |
| | FT | ResNet 4-bit | $15.65_{\pm4.52}$ | **100.0** | $135854.50_{\pm21399.94}$ | VGG 4-bit | $17.76_{\pm1.71}$ | **100.0** | $1900751.70_{\pm37329.44}$ |
| | TBT | | $66.36_{\pm0.07}$ | 99.8 | $271.24_{\pm15.98}$ | | $71.18_{\pm0.03}$ | **100.0** | $3231.00_{\pm345.68}$ |
| | T-BFA | | $65.86_{\pm0.42}$ | 80.4 | $24.79_{\pm19.02}$ | | $71.49_{\pm0.15}$ | 84.3 | $350.33_{\pm158.57}$ |
| | FSA | ACC: 66.77% | $66.44_{\pm0.21}$ | 99.9 | $157.53_{\pm33.66}$ | ACC: 71.76% | $71.69_{\pm0.09}$ | **100.0** | $441.32_{\pm111.26}$ |
| | GDA | | $66.54_{\pm0.22}$ | **100.0** | $\underline{11.45_{\pm3.82}}$ | | $\mathbf{71.73_{\pm0.03}}$ | **100.0** | $\underline{107.18_{\pm28.70}}$ |
| | **TA-LBF** | | $\mathbf{66.69_{\pm0.07}}$ | **100.0** | $\mathbf{7.96_{\pm2.50}}$ | | $\mathbf{71.73_{\pm0.03}}$ | **100.0** | $\mathbf{69.72_{\pm18.84}}$ |

**Results on ImageNet.** The results on ImageNet are shown in Table 1. It can be observed that GDA shows very competitive performance compared to other methods. However, our method obtains the highest PA-ACC, the fewest bit-flips (less than 8), and a 100% ASR in attacking ResNet. For VGG, our method also achieves a 100% ASR with the fewest $N_{flip}$ for both bit-widths. The $N_{flip}$ results of our method are mainly attributed to the cardinality constraint on the number of bit-flips. Moreover, for our method, the average PA-ACC degradation over four cases on ImageNet is only 0.06%, which demonstrates the stealthiness of our attack. When comparing the results of ResNet and VGG, an interesting observation is that all methods require significantly more bit-flips for VGG. One reason is that VGG is much wider than ResNet. Similar to the claim in (He et al., 2020), increasing the network width contributes to the robustness against the bit-flip based attack.

## 4.3 RESISTANCE TO DEFENSE METHODS

**Resistance to Piece-wise Clustering.** He et al. (2020) proposed a novel training technique, called piece-wise clustering, to enhance the network robustness against the bit-flip based attack. Such a training technique introduces an additional weight penalty to the inference loss, which has the effect of eliminating close-to-zero weights (He et al., 2020). We test the resistance of all attack methods to the piece-wise clustering. We conduct experiments with the 8-bit quantized ResNet on CIFAR-10 and ImageNet. Following the ideal configuration in (He et al., 2020), the clustering coefficient, which is a hyper-parameter of piece-wise clustering, is set to 0.001 in our evaluation. For our method, the initial $k$ is set to 50 on ImageNet and the rest settings are the same as those in Section 4.1. Besides the three metrics in Section 4.1, we also present the number of increased $N_{flip}$ compared to the model without defense (i.e., results in Table 1), denoted as $\Delta N_{flip}$.

The results of the resistance to the piece-wise clustering of all attack methods are shown in Table 2. It shows that the model trained with piece-wise clustering can improve the number of required bit-flips for all attack methods. However, our method still achieves a 100% ASR with the least number of bit-flips on both two datasets. Although TBT achieves a smaller $\Delta N_{flip}$ than ours on CIFAR-10, its ASR is only 52.3%, which also verifies the defense effectiveness of the piece-wise clustering. Compared with other methods, TA-LBF achieves the fewest $\Delta N_{flip}$ on ImageNet and the best PA-ACC on both datasets. These results demonstrate the superiority of our method over other methods when attacking models trained with piece-wise clustering.

Table 2: Results of all attack methods against the models with defense on CIFAR-10 and ImageNet (bold: the best; underline: the second best). The mean and standard deviation of PA-ACC and $N_{flip}$ are calculated by attacking the 1,000 images. Our method is denoted as **TA-LBF**. $\Delta N_{flip}$ denotes the increased $N_{flip}$ compared to the corresponding result in Table 1.

| Defense | Dataset | Method | ACC (%) | PA-ACC (%) | ASR (%) | $N_{flip}$ | $\Delta N_{flip}$ |
|---|---|---|---|---|---|---|---|
| Piece-wise Clustering | CIFAR-10 | FT | 91.01 | $84.06_{\pm3.56}$ | 99.5 | $1893.55_{\pm68.98}$ | 386.04 |
| | | TBT | | $87.05_{\pm1.69}$ | 52.3 | $254.20_{\pm10.22}$ | **7.50** |
| | | T-BFA | | $85.82_{\pm1.89}$ | 98.6 | $\underline{45.51_{\pm9.47}}$ | 35.60 |
| | | FSA | | $86.61_{\pm2.51}$ | 98.6 | $246.11_{\pm75.36}$ | 60.60 |
| | | GDA | | $84.12_{\pm4.77}$ | **100.0** | $52.76_{\pm16.29}$ | 25.93 |
| | | **TA-LBF** | | $\mathbf{87.30_{\pm2.74}}$ | **100.0** | $\mathbf{18.93_{\pm7.11}}$ | $\underline{13.36}$ |
| | ImageNet | FT | 63.62 | $43.44_{\pm2.07}$ | 92.2 | $762267.56_{\pm52179.46}$ | 484843.27 |
| | | TBT | | $63.07_{\pm0.04}$ | 81.8 | $1184.14_{\pm30.30}$ | 606.74 |
| | | T-BFA | | $62.82_{\pm0.27}$ | 90.1 | $273.56_{\pm191.29}$ | 248.99 |
| | | FSA | | $\underline{63.26_{\pm0.21}}$ | 99.5 | $729.94_{\pm491.83}$ | 288.73 |
| | | GDA | | $63.14_{\pm0.48}$ | **100.0** | $\underline{107.59_{\pm31.15}}$ | $\underline{89.05}$ |
| | | **TA-LBF** | | $\mathbf{63.52_{\pm0.14}}$ | **100.0** | $\mathbf{51.11_{\pm4.33}}$ | **43.74** |
| Larger Model Capacity | CIFAR-10 | FT | 94.29 | $86.46_{\pm2.84}$ | **100.0** | $2753.43_{\pm188.27}$ | 1245.92 |
| | | TBT | | $89.72_{\pm2.99}$ | 89.5 | $366.90_{\pm12.09}$ | 120.20 |
| | | T-BFA | | $\mathbf{91.16_{\pm1.42}}$ | 98.7 | $\underline{17.91_{\pm4.64}}$ | $\underline{8.00}$ |
| | | FSA | | $90.70_{\pm2.37}$ | 98.5 | $271.27_{\pm65.18}$ | 85.76 |
| | | GDA | | $89.83_{\pm3.02}$ | **100.0** | $48.96_{\pm21.03}$ | 22.13 |
| | | **TA-LBF** | | $\underline{90.96_{\pm2.63}}$ | **100.0** | $\mathbf{8.79_{\pm2.44}}$ | **3.22** |
| | ImageNet | FT | 71.35 | $63.51_{\pm1.29}$ | **100.0** | $507456.61_{\pm34517.04}$ | 230032.32 |
| | | TBT | | $71.12_{\pm0.04}$ | 99.9 | $1138.34_{\pm44.23}$ | 560.94 |
| | | T-BFA | | $70.84_{\pm0.30}$ | 88.9 | $40.23_{\pm27.29}$ | 15.66 |
| | | FSA | | $\mathbf{71.30_{\pm0.04}}$ | **100.0** | $449.70_{\pm106.42}$ | 8.49 |
| | | GDA | | $\mathbf{71.30_{\pm0.05}}$ | **100.0** | $\underline{20.01_{\pm6.04}}$ | $\underline{1.47}$ |
| | | **TA-LBF** | | $\mathbf{71.30_{\pm0.04}}$ | **100.0** | $\mathbf{8.48_{\pm2.52}}$ | **1.11** |

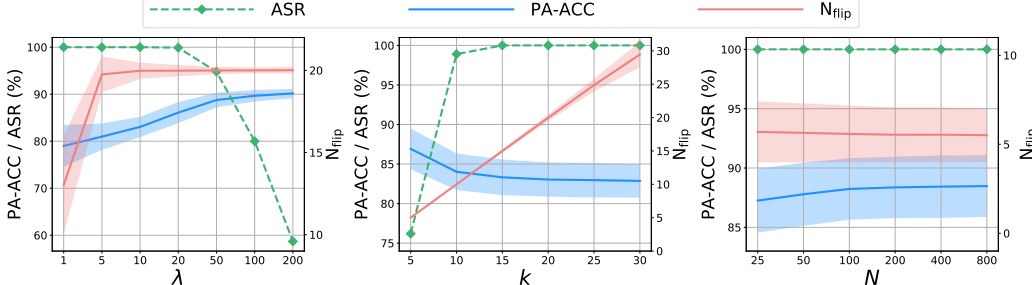

Figure 2: Results of TA-LBF with different parameters $\lambda$, $k$, and the number of auxiliary samples $N$ on CIFAR-10. Regions in shadow indicate the standard deviation of attacking the 1,000 images.

**Resistance to Larger Model Capacity.** Previous studies (He et al., 2020; Rakin et al., 2020b) observed that increasing the network capacity can improve the robustness against the bit-flip based attack. Accordingly, we evaluate all attack methods against the models with a larger capacity using the 8-bit quantized ResNet on both datasets. Similar to the strategy in (He et al., 2020), we increase the model capacity by varying the network width (*i.e.*, 2× width in our experiments). All settings of our method are the same as those used in Section 4.1.

The results are presented in Table 2. We observe that all methods require more bit-flips to attack the model with the 2× width. To some extent, it demonstrates that the wider network with the same architecture is more robust against the bit-flip based attack. However, our method still achieves a 100% ASR with the fewest $N_{flip}$ and $\Delta N_{flip}$. Moreover, when comparing the two defense methods, we find that piece-wise clustering performs better than the model with a larger capacity in terms of $\Delta N_{flip}$. However, piece-wise clustering training also causes the accuracy decrease of the original model (*e.g.*, from 92.16% to 91.01% on CIFAR-10). We provide more results in attacking models with defense under different settings in **Appendix F**.

Figure 3: Visualization of decision boundaries of the original model and the post attack models. The attacked sample from Class 3 is misclassified into the Class 1 by FSA, GDA, and our method.

## 4.4 Ablation Study

We perform ablation studies on parameters $\lambda$ and $k$, and the number of auxiliary samples $N$. We use the 8-bit quantized ResNet on CIFAR-10 as the representative for analysis. We discuss the attack performance of TA-LBF under different values of $\lambda$ while $k$ is fixed at 20, and under different values of $k$ while $\lambda$ is fixed at 10. To analyze the effect of $N$, we configure $N$ from 25 to 800 and keep other settings the same as those in Section 4.1. The results are presented in Fig. 2. We observe that our method achieves a 100% ASR when $\lambda$ is less than 20. As expected, the PA-ACC increases while the ASR decreases along with the increase of $\lambda$. The plot of parameter $k$ presents that $k$ can exactly limit the number of bit-flips, while other attack methods do not involve such constraint. This advantage is critical since it allows the attacker to identify limited bits to perform an attack when the budget is fixed. As shown in the figure, the number of auxiliary samples less than 200 has a marked positive impact on the PA-ACC. It's intuitive that more auxiliary samples can lead to a better PA-ACC. The observation also indicates that TA-LBF still works well without too many auxiliary samples.

## 4.5 Visualization of Decision Boundary

To further compare FSA and GDA with our method, we visualize the decision boundaries of the original and the post attack models in Fig. 3. We adopt a four-layer Multi-Layer Perceptron trained with the simulated 2-D Blob dataset from 4 classes. The original decision boundary indicates that the original model classifies all data points almost perfectly. The attacked sample is classified into Class 3 by all methods. Visually, GDA modifies the decision boundary drastically, especially for Class 0. However, our method modifies the decision boundary mainly around the attacked sample. Althoug FSA is comparable to ours visually in Fig. 3, it flips $10\times$ bits than GDA and TA-LBF. In terms of the numerical results, TA-LBF achieves the best PA-ACC and the fewest $N_{flip}$. This finding verifies that our method can achieve a successful attack even only tweaking the original classifier.

## 5 Conclusion

In this work, we have presented a novel attack paradigm that the weights of a deployed DNN can be slightly changed via bit flipping in the memory, to give a target prediction for a specific sample, while the predictions on other samples are not significantly influenced. Since the weights are stored as binary bits in the memory, we formulate this attack as a binary integer programming (BIP) problem, which can be effectively and efficiently solved by a continuous algorithm. Since the critical bits are determined through optimization, the proposed method can achieve the attack goals by flipping a few bits, and it shows very good performance under different experimental settings.

## Acknowledgments

This work is supported in part by the National Key Research and Development Program of China under Grant 2018YFB1800204, the National Natural Science Foundation of China under Grant 61771273, the R&D Program of Shenzhen under Grant JCYJ20180508152204044. Baoyuan Wu is supported by the Natural Science Foundation of China under grant No. 62076213, and the university development fund of the Chinese University of Hong Kong, Shenzhen under grant No. 01001810.

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

# A  UPDATE $\hat{b}$ BY GRADIENT DESCENT

In this section, we derive the gradient of $L$ *w.r.t.* $\hat{b}$, which is adopted to update $\hat{b}^{r+1}$ by gradient descent (see Eq. (10)). The derivation consists of the following parts.

**Derivation of $\partial\mathcal{L}_1(\hat{b})/\partial\hat{b}$.**  For clarity, here we firstly repeat some definitions,

$$\mathcal{L}_1(\hat{\mathbf{B}}_s,\hat{\mathbf{B}}_t) = \max\big(m - p(\boldsymbol{x};\boldsymbol{\Theta},\hat{\mathbf{B}}_t) + \delta, 0\big) + \max\big(p(\boldsymbol{x};\boldsymbol{\Theta},\hat{\mathbf{B}}_s) - m + \delta, 0\big), \qquad (12)$$

$$p(\boldsymbol{x};\boldsymbol{\Theta},\hat{\mathbf{B}}_i) = [h(\hat{\mathbf{B}}_{i,1}); h(\hat{\mathbf{B}}_{i,2}); ...; h(\hat{\mathbf{B}}_{i,C})]^\top g(\boldsymbol{x};\boldsymbol{\Theta}), \qquad (13)$$

$$h(\boldsymbol{v}) = (-2^{Q-1} \cdot v_Q + \sum_{i=1}^{Q-1} 2^{i-1} \cdot v_i) \cdot \Delta^l. \qquad (14)$$

Then, we obtain that

$$\frac{\partial p(\boldsymbol{x};\boldsymbol{\Theta},\hat{\mathbf{B}}_i)}{\partial\hat{\mathbf{B}}_i} = [g_1(\boldsymbol{x};\boldsymbol{\Theta}) \cdot (\frac{\nabla h(\hat{\mathbf{B}}_{i,1})}{\partial\hat{\mathbf{B}}_{i,1}})^\top; ...; g_C(\boldsymbol{x};\boldsymbol{\Theta}) \cdot (\frac{\nabla h(\hat{\mathbf{B}}_{i,C})}{\nabla\hat{\mathbf{B}}_{i,C}})^\top], \qquad (15)$$

where $\frac{\nabla h(\boldsymbol{v})}{\nabla\boldsymbol{v}} = [2^0; 2^1, \ldots, 2^{Q-2}; -2^{Q-1}] \cdot \Delta^l$ is a constant, and here $l$ indicates the last layer; $g_j(\boldsymbol{x};\boldsymbol{\Theta})$ denotes the $j$-th entry of the vector $g(\boldsymbol{x};\boldsymbol{\Theta})$. Utilizing (15), we have

$$\frac{\partial\mathcal{L}_1(\hat{\mathbf{B}}_s,\hat{\mathbf{B}}_t)}{\partial\hat{\mathbf{B}}_s} = \begin{cases} \frac{\partial p(\boldsymbol{x};\boldsymbol{\Theta},\hat{\mathbf{B}}_s)}{\partial\hat{\mathbf{B}}_s}, & \text{if } p(\boldsymbol{x};\boldsymbol{\Theta},\mathbf{B}_s) > m - \delta \\ \mathbf{0}, & \text{otherwise} \end{cases},$$
$$\frac{\partial\mathcal{L}_1(\hat{\mathbf{B}}_s,\hat{\mathbf{B}}_t)}{\partial\hat{\mathbf{B}}_t} = \begin{cases} -\frac{\partial p(\boldsymbol{x};\boldsymbol{\Theta},\hat{\mathbf{B}}_t)}{\partial\hat{\mathbf{B}}_t}, & \text{if } p(\boldsymbol{x};\boldsymbol{\Theta},\mathbf{B}_t) < m + \delta \\ \mathbf{0}, & \text{otherwise} \end{cases}. \qquad (16)$$

Thus, we obtain that

$$\frac{\partial\mathcal{L}_1(\hat{b})}{\partial\hat{b}} = \big[\text{Reshape}\big(\frac{\partial\mathcal{L}_1(\hat{\mathbf{B}}_s,\hat{\mathbf{B}}_t)}{\partial\hat{\mathbf{B}}_s}\big); \text{Reshape}\big(\frac{\partial\mathcal{L}_1(\hat{\mathbf{B}}_s,\hat{\mathbf{B}}_t)}{\partial\hat{\mathbf{B}}_t}\big)\big], \qquad (17)$$

where $\text{Reshape}(\cdot)$ elongates a matrix to a vector along the column.

**Derivation of $\partial\mathcal{L}_2(\hat{b})/\partial\hat{b}$.**  For clarity, here we firstly repeat the following definition

$$\mathcal{L}_2(\hat{\mathbf{B}}_s,\hat{\mathbf{B}}_t) = \sum_{i=1}^{N} \ell\big(f(\boldsymbol{x}_i;\boldsymbol{\Theta},\mathbf{B}_{\{1,...,K\}\setminus\{s,t\}},\hat{\mathbf{B}}_s,\hat{\mathbf{B}}_t), y_i\big), \qquad (18)$$

where $f_j(\boldsymbol{x}_i;\boldsymbol{\Theta},\mathbf{B}_{\{1,...,K\}\setminus\{s,t\}},\hat{\mathbf{B}}_s,\hat{\mathbf{B}}_t) = \text{Softmax}(p(\boldsymbol{x}_i;\boldsymbol{\Theta},\hat{\mathbf{B}}_j))$ or $\text{Softmax}(p(\boldsymbol{x}_i;\boldsymbol{\Theta},\mathbf{B}_j))$ indicates the posterior probability of $\boldsymbol{x}_i$ *w.r.t.* class $j$, and we simply denote $f(\boldsymbol{x}_i) \in [0,1]^K$ as the posterior probability vector of $\boldsymbol{x}_i$. $\{(\boldsymbol{x}_i,y_i)\}_{i=1}^N$ denotes the auxiliary sample set. $\ell(\cdot,\cdot)$ is specified as the cross entropy loss. Then, we have

$$\frac{\partial\mathcal{L}_2(\hat{\mathbf{B}}_s,\hat{\mathbf{B}}_t)}{\partial\hat{\mathbf{B}}_s} = \sum_{i=1}^{N}\Big[\big(\mathbb{I}(y_i = s) - f_s(\boldsymbol{x}_i;\boldsymbol{\Theta},\mathbf{B}_{\{1,...,K\}\setminus\{s,t\}},\hat{\mathbf{B}}_s,\hat{\mathbf{B}}_t)\big) \cdot \frac{\partial p(\boldsymbol{x}_i;\boldsymbol{\Theta},\hat{\mathbf{B}}_s)}{\partial\hat{\mathbf{B}}_s}\Big], \quad (19)$$

$$\frac{\partial\mathcal{L}_2(\hat{\mathbf{B}}_s,\hat{\mathbf{B}}_t)}{\partial\hat{\mathbf{B}}_t} = \sum_{i=1}^{N}\Big[\big(\mathbb{I}(y_i = t) - f_t(\boldsymbol{x}_i;\boldsymbol{\Theta},\mathbf{B}_{\{1,...,K\}\setminus\{s,t\}},\hat{\mathbf{B}}_s,\hat{\mathbf{B}}_t)\big) \cdot \frac{\partial p(\boldsymbol{x}_i;\boldsymbol{\Theta},\hat{\mathbf{B}}_t)}{\partial\hat{\mathbf{B}}_t}\Big], \quad (20)$$

where $\mathbb{I}(a) = 1$ of $a$ is true, otherwise $\mathbb{I}(a) = 0$. Thus, we obtain

$$\frac{\partial\mathcal{L}_2(\hat{b})}{\partial\hat{b}} = \Big[\text{Reshape}\big(\frac{\partial\mathcal{L}_2(\hat{\mathbf{B}}_s,\hat{\mathbf{B}}_t)}{\partial\hat{\mathbf{B}}_s}\big); \text{Reshape}\big(\frac{\partial\mathcal{L}_2(\hat{\mathbf{B}}_s,\hat{\mathbf{B}}_t)}{\partial\hat{\mathbf{B}}_t}\big)\Big]. \qquad (21)$$

**Derivation of $\partial L(\hat{b})/\partial\hat{b}$.**  According to Eq. (8), and utilizing Eqs. (17) and (21), we obtain

$$\frac{\partial L(\hat{b})}{\partial\hat{b}} = \frac{\partial\mathcal{L}_1(\hat{b})}{\partial\hat{b}} + \frac{\partial\mathcal{L}_2(\hat{b})}{\partial\hat{b}} + \boldsymbol{z}_1 + \boldsymbol{z}_2 + \rho_1(\hat{b} - \boldsymbol{u}_1) + \rho_2(\hat{b} - \boldsymbol{u}_2) + 2(\hat{b} - b) \cdot [\boldsymbol{z}_3 + \rho_3||\hat{b} - b||_2^2 - k + u_3]. \qquad (22)$$

## B  ALGORITHM OUTLINE

---

**Algorithm 1** Continuous optimization for the BIP problem (5).

---

**Input:** The original quantized DNN model $f$ with weights $\Theta, \mathbf{B}$, attacked sample $\boldsymbol{x}$ with ground-truth label $s$, target class $t$, auxiliary sample set $\{(\boldsymbol{x}_i, y_i)\}_{i=1}^{N}$, hyper-parameters $\lambda$, $k$, and $\delta$.

**Output:** $\hat{b}$.

1: Initial $\boldsymbol{u}_1^0, \boldsymbol{u}_2^0, u_3^0, \boldsymbol{z}_1^0, \boldsymbol{z}_2^0, z_3^0, \hat{\boldsymbol{b}}^0$ and let $r \leftarrow 0$;
2: **while** not converged **do**
3:     Update $\boldsymbol{u}_1^{r+1}, \boldsymbol{u}_2^{r+1}$ and $u_3^{r+1}$ as Eq. (9);
4:     Update $\hat{\boldsymbol{b}}^{r+1}$ as Eq. (10);
5:     Update $\boldsymbol{z}_1^{r+1}, \boldsymbol{z}_2^{r+1}$ and $z_3^{r+1}$ as Eq. (11);
6:     $r \leftarrow r + 1$.
7: **end while**

---

## C  COMPLEXITY ANALYSIS

Table 3: Running time (seconds) of attacking one image for different methods. The mean and standard deviation are calculated by 10 attacks.

|  | FT | TBT | T-BFA | FSA | GDA | TA-LBF |
|---|---|---|---|---|---|---|
| CIFAR-10 | $15.54_{\pm 1.64}$ | $389.12_{\pm 27.79}$ | $35.05_{\pm 15.79}$ | $2.71_{\pm 0.48}$ | $0.67_{\pm 0.54}$ | $113.38_{\pm 6.54}$ |
| ImageNet | $124.32_{\pm 3.61}$ | $31425.81_{\pm 540.60}$ | $19.16_{\pm 3.52}$ | $65.28_{\pm 2.49}$ | $61.97_{\pm 1.59}$ | $222.95_{\pm 9.39}$ |

The computational complexity of the proposed algorithm (*i.e.*, Algorithm 1) consists of two parts, the forward and backward pass. In terms of the forward pass, since $\Theta, \mathbf{B}_{\{1,...,K\}\setminus\{s,t\}}$ are fixed during the optimization, their involved terms, including $g(\boldsymbol{x}; \Theta)$ and $p(\boldsymbol{x}; \Theta, \mathbf{B}_i)|_{i \neq s,t}$, are calculated only one time. The main cost from $\hat{\mathbf{B}}_s$ and $\hat{\mathbf{B}}_t$ is $O(2(N+1)C^2Q)$ per iteration, as there are $N+1$ samples. In terms of the backward pass, the main cost is from the update of $\hat{\boldsymbol{b}}^{r+1}$, which is $O(2(N+1)CQ)$ per iteration in the gradient descent. Since all other updates are very simple, their costs are omitted here. Thus, the overall computational cost is $O\big(T_{outer}[2(N+1)CQ \cdot (C + T_{inner})]\big)$, with $T_{outer}$ being the iteration of the overall algorithm and $T_{inner}$ indicating the number of gradient steps in updating $\hat{\boldsymbol{b}}^{r+1}$. As shown in Section D, the proposed method TA-LBF always converges very fast in our experiments, thus $T_{outer}$ is not very large. As demonstrated in Section E.3, $T_{inner}$ is set to 5 in our experiments. In short, the proposed method can be optimized very efficiently.

Besides, we also compare the computational complexity of different attacks empirically. Specifically, we compare the running time of attacking one image of different methods against the 8-bit quantized ResNet on CIFAR-10 and ImageNet dataset. As shown in Table 3, TBT is the most time-consuming method among all attacks. Although the proposed TA-LBF is not superior to T-BFA, FSA, and GDA in running time, this gap can be tolerated when attacking a single image in the deployment stage. Besides, our method performs better in terms of PA-ACC, ASR, and $N_{\text{flip}}$ as demonstrated in our experiments.

## D  NUMERICAL CONVERGENCE ANALYSIS

We present the numerical convergence of TA-LBF in Fig. 4. Note that $||\hat{\boldsymbol{b}} - \boldsymbol{u}_1||_2^2$ and $||\hat{\boldsymbol{b}} - \boldsymbol{u}_2||_2^2$ characterize the degree of satisfaction of the box and $\ell_2$-sphere constraint, respectively. For the two examples of CIFAR-10 and ImageNet, the values of both indicators first increase, then drop, and finally close to 0. Another interesting observation is that $\mathcal{L}_1 + \lambda\mathcal{L}_2$ first decreases evidently and then increases slightly. Such findings illustrate the optimization process of TA-LBF. In the early iterations, modifying the model parameters tends to achieve the two goals mentioned in Section 3.1; in the late iterations, $\hat{\boldsymbol{b}}$ is encouraged to satisfy the box and $l_2$-sphere constraint. We also observe that both examples stop when meeting $||\hat{\boldsymbol{b}} - \boldsymbol{u}_1||_2^2 \leq 10^{-4}$ and $||\hat{\boldsymbol{b}} - \boldsymbol{u}_2||_2^2 \leq 10^{-4}$ and do not

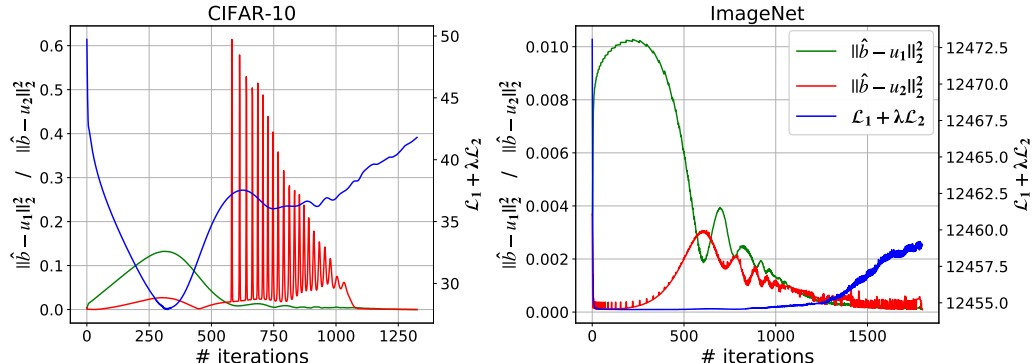

Figure 4: Numerical convergence analysis of TA-LBF *w.r.t.* the attacked sample on CIFAR-10 and ImageNet, respectively. We present the values of $||\hat{\boldsymbol{b}} - \boldsymbol{u}_1||_2^2$, $||\hat{\boldsymbol{b}} - \boldsymbol{u}_2||_2^2$ and $\mathcal{L}_1 + \lambda\mathcal{L}_2$ at different iterations in attacking 8-bit quantized ResNet. Note that $\lambda$ in the left figure is 100 and $\lambda$ in the right figure is $10^4$.

exceed the maximum number of iterations (*i.e.*, 2000). The numerical results demonstrate the fast convergence of our method in practice.

## E   EVALUATION SETUP

### E.1   BASELINE METHODS

Since GDA (Liu et al., 2017a) and FSA (Zhao et al., 2019) are originally designed for attacking the full-precision network, we adapt these two methods to attack the quantized network by applying quantization-aware training (Jacob et al., 2018). We adopt the $\ell_0$-norm for FSA (Liu et al., 2017a) and modification compression for GDA (Zhao et al., 2019) to reduce the number of the modified parameters. Among three types of T-BFA (Rakin et al., 2020b), we compare to the most comparable method: the 1-to-1 stealthy attack scheme. The purpose of this attack scheme is to misclassify samples of a single source class into the target class while maintaining the prediction accuracy of other samples. Besides, we take the fine-tuning (FT) of the last fully-connected layer as a basic attack and present its results. We perform attack once for each selected image except TBT (Rakin et al., 2020a) and totally 1,000 attacks on each dataset. The attack objective of TBT is that the attacked DNN model misclassifies all inputs with a trigger to a certain target class. Due to such objective, the number of attacks for TBT is equal to the number of target classes (*i.e.*, 10 attacks on CIFAR-10 and 50 attacks on ImageNet).

### E.2   TARGET MODELS

According to the setting in (Rakin et al., 2020a;b), we adopt two popular network architectures: ResNet (He et al., 2016) and VGG (Simonyan & Zisserman, 2015) for evaluation. On CIFAR-10, we perform experiments on ResNet-20 and VGG-16. On ImageNet, we use the pre-trained ResNet-18[*] and VGG-16[†] network. We quantize all networks to the 4-bit and 8-bit quantization level using the layer-wise uniform weight quantization scheme, which is similar to the one involved in the Tensor-RT solution (Migacz, 2017).

### E.3   PARAMETER SETTINGS OF TA-LBF

For each attack, we adopt a strategy for jointly searching $\lambda$ and $k$. Specifically, for an initially given $k$, we search $\lambda$ from a relatively large initial value and divide it by 2 if the attack does not succeed. The maximum search times of $\lambda$ for a fixed $k$ is set to 8. If it exceeds the maximum search times,

---

[*]Downloaded from `https://download.pytorch.org/models/resnet18-5c106cde.pth`
[†]Downloaded from `https://download.pytorch.org/models/vgg16_bn-6c64b313.pth`

Table 4: Results of all attack methods against models trained with piece-wise clustering on CIFAR-10 (bold: the best; underline: the second best). We adopt different clustering coefficients, including 0.0005, 0.005, and 0.01. The mean and standard deviation of PA-ACC and $N_{flip}$ are calculated by attacking the 1,000 images. Our method is denoted as **TA-LBF**. $\Delta N_{flip}$ denotes the increased $N_{flip}$ compared to the corresponding result in Table 1.

| Clustering Coefficient | Method | ACC (%) | PA-ACC (%) | ASR (%) | $N_{flip}$ | $\Delta N_{flip}$ |
|---|---|---|---|---|---|---|
| 0.0005 | FT | 91.42 | $84.28_{\pm 3.49}$ | **100.0** | $1868.26_{\pm 72.48}$ | 360.75 |
| | TBT | | $\mathbf{87.97}_{\pm \mathbf{1.75}}$ | 66.1 | $250.30_{\pm 10.97}$ | **3.60** |
| | T-BFA | | $86.20_{\pm 1.96}$ | 98.5 | $\underline{30.95}_{\pm 6.50}$ | 21.04 |
| | FSA | | $87.17_{\pm 2.44}$ | 98.5 | $222.70_{\pm 56.52}$ | 37.19 |
| | GDA | | $85.28_{\pm 4.16}$ | **100.0** | $41.33_{\pm 12.84}$ | 14.50 |
| | **TA-LBF** | | $\underline{87.92}_{\pm 2.54}$ | **100.0** | $\mathbf{13.47}_{\pm \mathbf{5.34}}$ | $\underline{7.90}$ |
| 0.005 | FT | 88.03 | $81.08_{\pm 3.61}$ | 97.9 | $1774.69_{\pm 51.47}$ | 267.18 |
| | TBT | | $82.96_{\pm 2.18}$ | 12.7 | $246.80_{\pm 16.06}$ | **0.10** |
| | T-BFA | | $80.80_{\pm 2.64}$ | 98.1 | $\underline{61.72}_{\pm 12.17}$ | 51.81 |
| | FSA | | $\underline{83.10}_{\pm 2.75}$ | 98.4 | $231.66_{\pm 89.21}$ | 46.15 |
| | GDA | | $79.23_{\pm 6.25}$ | $\underline{99.9}$ | $64.87_{\pm 22.78}$ | 38.04 |
| | **TA-LBF** | | $\mathbf{83.63}_{\pm \mathbf{3.47}}$ | **100.0** | $\mathbf{25.52}_{\pm \mathbf{11.59}}$ | $\underline{19.95}$ |
| 0.01 | FT | 85.65 | $78.73_{\pm 3.54}$ | 98.3 | $1748.54_{\pm 46.19}$ | 241.03 |
| | TBT | | $79.86_{\pm 2.04}$ | 10.1 | $236.50_{\pm 10.93}$ | **-10.20** |
| | T-BFA | | $76.67_{\pm 3.41}$ | 98.1 | $\underline{55.49}_{\pm 11.77}$ | 45.58 |
| | FSA | | $\underline{80.45}_{\pm 3.14}$ | 98.0 | $220.28_{\pm 101.01}$ | 34.77 |
| | GDA | | $75.33_{\pm 7.83}$ | $\underline{99.7}$ | $59.17_{\pm 23.63}$ | 32.34 |
| | **TA-LBF** | | $\mathbf{80.51}_{\pm \mathbf{4.39}}$ | **100.0** | $\mathbf{24.60}_{\pm \mathbf{13.03}}$ | $\underline{19.03}$ |

we double $k$ and search $\lambda$ from the relatively large initial value. The maximum search times of $k$ is set to 4. On CIFAR-10, the initial $k$ and $\lambda$ are set to 5 and 100. On ImageNet, $\lambda$ is initialized as $10^4$; $k$ is initialized as 5 and 50 for ResNet and VGG, respectively. On CIFAR-10, the $\delta$ in $\mathcal{L}_1$ is set to 10. On ImageNet, the $\delta$ is set to 3 and increased to 10 if the attack fails. $\boldsymbol{u}_1$ and $\boldsymbol{u}_2$ are initialized as $\boldsymbol{b}$ and $u_3$ is initialized as 0. $\boldsymbol{z}_1$ and $\boldsymbol{z}_2$ are initialized as $\boldsymbol{0}$ and $z_3$ is initialized as 0. $\hat{\boldsymbol{b}}$ is initialized as $\boldsymbol{b}$. During each iteration, the number of gradient steps for updating $\hat{\boldsymbol{b}}$ is 5 and the step size is set to 0.01 on both datasets. Hyper-parameters $(\rho_1, \rho_2, \rho_3)$ (see Eq. (11)) are initialized as $(10^{-4}, 10^{-4}, 10^{-5})$ on both datasets, and increase by $\rho_i \leftarrow \rho_i \times 1.01$, $i = 1, 2, 3$ after each iteration. The maximum values of $(\rho_1, \rho_2, \rho_3)$ are set to $(50, 50, 5)$ on both datasets. Besides the maximum number of iterations (*i.e.*, 2000), we also set another stopping criterion, *i.e.*, $||\hat{\boldsymbol{b}} - \boldsymbol{u}_1||_2^2 \leq 10^{-4}$ and $||\hat{\boldsymbol{b}} - \boldsymbol{u}_2||_2^2 \leq 10^{-4}$.

# F    MORE RESULTS ON RESISTANCE TO DEFENSE METHODS

## F.1    RESISTANCE TO PIECE-WISE CLUSTERING

We conduct experiments using the 8-bit quantized ResNet on CIFAR-10 with different clustering coefficients. We set the maximum search times of $k$ to 5 for clustering coefficient 0.005 and 0.01 and keep the rest settings the same as those in Section 4.1. The results are presented in Table 4. As shown in the table, all values of $N_{flip}$ are larger than attacking models without defense for all methods, which is similar to Table 2. Our method achieves a 100% ASR with the fewest $N_{flip}$ under the three clustering coefficients. Although TBT obtains a smaller $\Delta N_{flip}$ than our method, it fails to achieve a satisfactory ASR. For example, TBT achieves only a 10.1% ASR when the clustering coefficient is set to 0.01. We observe that for all clustering coefficients, piece-wise clustering reduces the original accuracy. Such a phenomenon is more significant as the clustering coefficient increases. The results also show that there is no guarantee that if the clustering coefficient is larger (*e.g.*, 0.01), the model is more robust, which is consistent with the finding in (He et al., 2020).

Table 5: Results of all attack methods against models with a larger capacity on CIFAR-10. We adopt $3\times$ and $4\times$ width networks. The mean and standard deviation of PA-ACC and $N_{flip}$ are calculated by attacking the 1,000 images. $\Delta N_{flip}$ denotes the increased $N_{flip}$ compared to the corresponding result in Table 1.

| Model Width | Method | ACC (%) | PA-ACC (%) | ASR (%) | $N_{flip}$ | $\Delta N_{flip}$ |
|---|---|---|---|---|---|---|
| | FT | | $86.96_{\pm2.79}$ | **100.0** | $4002.52_{\pm281.24}$ | 2495.01 |
| | TBT | | $90.67_{\pm5.23}$ | 74.1 | $504.70_{\pm20.44}$ | 258.00 |
| $3\times$ | T-BFA | 94.90 | $\mathbf{92.18}_{\pm1.14}$ | 98.9 | $\underline{30.50_{\pm7.52}}$ | $\underline{20.59}$ |
| | FSA | | $91.40_{\pm2.38}$ | 99.0 | $342.20_{\pm79.44}$ | 156.69 |
| | GDA | | $90.79_{\pm2.91}$ | **100.0** | $67.53_{\pm27.45}$ | 40.70 |
| | **TA-LBF** | | $\underline{91.42_{\pm2.81}}$ | **100.0** | $\mathbf{12.29}_{\pm4.18}$ | **6.72** |
| | FT | | $86.94_{\pm2.78}$ | **100.0** | $4527.68_{\pm369.35}$ | 3020.17 |
| | TBT | | $85.39_{\pm5.08}$ | 90.1 | $625.50_{\pm32.38}$ | 378.80 |
| $4\times$ | T-BFA | 95.02 | $\mathbf{92.49}_{\pm1.22}$ | 99.4 | $\underline{19.14_{\pm5.04}}$ | $\underline{9.23}$ |
| | FSA | | $91.60_{\pm2.42}$ | 98.7 | $338.93_{\pm100.12}$ | 153.42 |
| | GDA | | $90.76_{\pm3.00}$ | **100.0** | $66.92_{\pm40.32}$ | 40.09 |
| | **TA-LBF** | | $\underline{90.94_{\pm3.11}}$ | **100.0** | $\mathbf{8.37}_{\pm2.80}$ | **2.80** |

## F.2 RESISTANCE TO LARGER MODEL CAPACITY

Besides the results of networks with a $2\times$ width shown in Section 4.3, we also evaluate all methods against models with a $3\times$ and $4\times$ width. All settings are the same as those used in Section 4.1. The results are provided in Table 5. Among all attack methods, our method is least affected by increasing the network width. Especially for the network with a $4\times$ width, our $\Delta N_{flip}$ is only 2.80. The results demonstrate the superiority of the formulated BIP problem and optimization. Moreover, compared with piece-wise clustering, having a larger model capacity can improve the original accuracy, but increases the model size and the computation complexity.

## G DISCUSSIONS

### G.1 COMPARING BACKDOOR, ADVERSARIAL, AND WEIGHT ATTACK

An attacker can achieve malicious purposes utilizing backdoor, adversarial, and weight attacks. In this section, we emphasize the differences among them.

Backdoor attack happens in the training stage and requires that the attacker can tamper the training data even the training process (Liu et al., 2020b; Li et al., 2020). Through poisoning some training samples with a trigger, the attacker can control the behavior of the attacked DNN in the inference stage. For example, images with reflections are misclassified into a target class, while benign images are classified normally (Liu et al., 2020a). However, such an attack paradigm causes the accuracy degradation on benign samples, which makes it detectable for users. Besides, these methods also require to modify samples in the inference stage, which is sometimes impossible for the attacker. Many defense methods against backdoor attack have been proposed, such as the preprocessing-based defense (Liu et al., 2017b), the model reconstruction-based defense (Liu et al., 2018a), and the trigger synthesis-based defense (Wang et al., 2019).

Adversarial attack modifies samples in the inference stage by adding small perturbations that remain imperceptible to the human vision system (Akhtar & Mian, 2018). Since adversarial attack only modifies inputs while keeping the model unchanged, it has no effect on the benign samples. Besides the basic white-box attack, the black-box attack (Wu et al., 2020b; Chen et al., 2020) and universal attack (Zhang et al., 2020b;a) have attracted wide attention. Inspired by its success in the classification, it also has been extended to other tasks, including image captioning (Xu et al., 2019), retrieval (Bai et al., 2020; Feng et al., 2020), *etc.*. Similarly, recent studies have demonstrated many defense methods against adversarial attack, including the preprocessing-based defense (Xie et al., 2018), the detection-based defense (Xu et al., 2017), and the adversarial learning-based defense (Carmon et al., 2019; Wu et al., 2020c).

Weight attack modifies model parameters in the deployment stage, which is the studied paradigm in this work. Weight attack generally aims at misleading the DNN model on the selected sample(s), while having a minor effect on other samples (Zhao et al., 2019; Rakin et al., 2020b). Many studies (Yao et al., 2020; Breier et al., 2018; Pan, 2020) have demonstrated that the DNN parameters can be modified in the bit-level in memory using fault injection techniques (Agoyan et al., 2010; Kim et al., 2014; Selmke et al., 2015) in practice. Note that the defense methods against weight attack have been not well studied. Although some defense methods (He et al., 2020) were proposed, they cannot achieve satisfactory performance. For example, our method can still achieve a 100% attack success rate against two proposed defense methods. Our work would encourage further investigation on the security of the model parameters from both attack and defense sides.

## G.2 Comparing TA-LBF with Other Weight Attacks

We compare our TA-LBF with other weight attack methods, including TBT (Rakin et al., 2020a), T-BFA (Rakin et al., 2020b), GDA (Liu et al., 2017a), and FSA (Zhao et al., 2019) in this section. TBT tampers both the test sample and the model parameters. Specifically, it first locates critical bits and generates a trigger, and then flips these bits to classify all inputs embedded with the trigger to a target class. However, the malicious samples are easily detected by human inspection or many detection methods (Tran et al., 2018; Du et al., 2020). We do not modify the samples to perform TA-LBF, which makes the attack more stealthy. Rakin et al. (2020b) proposed T-BFA which misclassifies all samples (N-to-1 version) or samples from a source class (1-to-1 version) into a target class. Our method aims at misclassifying a specific sample, which meets the attacker's requirement in some scenarios. For example, the attacker wants to manipulate the behavior of a face recognition engine on a specific input. Since it affects multiple samples, T-BFA maybe not stealthy enough in attacking real-world applications. GDA (Liu et al., 2017a) and FSA (Zhao et al., 2019) modify model parameters at the weight-level rather than bit-level. They are designed for misclassifying multiple samples from arbitrary classes, which makes it infeasible for them to only modify the parameters connected to the source and target class. They modify more parameters than our method as shown in the experiments, it might be due to the reason discussed above. Besides, TBT, T-BFA, and GDA determine the critical weights to modify using heuristic strategies, while our TA-LBF adopts optimization-based methods. Although FSA applies ADMM for solving the optimization problem, it has no explicit constraint to control the number of modified parameters, which makes it intends to modify more parameters than GDA and our TA-LBF.

## H Trade-off between Three Evaluation Metrics

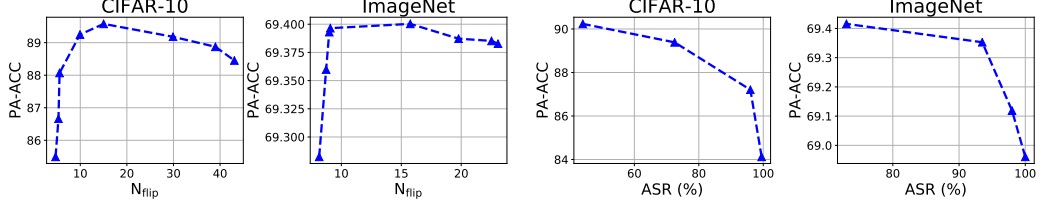

Figure 5: Curves of the trade-off between PA-ACC and $N_{flip}$ and the trade-off between PA-ACC and ASR for the proposed TA-LBF on two datasets.

In this section, we investigate the trade-off between three adopted evaluation metrics (*i.e.*, PA-ACC, ASR, and $N_{flip}$) for our attack. All experiments are conducted on CIFAR-10 and ImageNet dataset in attacking the 8-bit quantized ResNet.

We firstly discuss the trade-off between PA-ACC and $N_{flip}$ by fixing the ASR as 100% using the search strategy in Appendix E.3 and adjusting the initial $\lambda$ and $k$ to obtain different attack results. The two curves on the left show that increasing the $N_{flip}$ can improve the PA-ACC when $N_{flip}$ is relatively small; the PA-ACC decreases with the increase of $N_{flip}$ when $N_{flip}$ is greater than a threshold. This phenomenon demonstrates that constraining the number of bit-flips is essential to ensure the attack stealthiness, as mentioned in Section 3.2. To study the trade-off between PA-ACC

and ASR, we fix the parameter $k$ as 10 for approximately 10 bit-flips and adjust the parameter $\lambda$ to obtain different PA-ACC and ASR results. The trade-off curves between PA-ACC and ASR show that increasing ASR can decrease the PA-ACC significantly. Therefore, how to achieve high ASR and PA-ACC simultaneously is still an important open problem.

