# OpenReview forum: "Targeted Attack against Deep Neural Networks via Flipping Limited Weight Bits"
_ICLR.cc/2021/Conference — ICLR 2021 Poster_

### Official Review · AnonReviewer4 · 2020-10-28
**Unclear motivation and improvement over prior works**

**Rating:** 5
**Confidence:** 4

**Review:**

The paper proposes a bit-flip attack where model parameters weights are altered such that a certain sample is misclassified to a target class.  While the utilized optimization strategy and the combination of techniques seems interesting, a major concern is the motivation behind the proposed method and why it stands out against prior works.
- Comparisons in the experiment section show that prior methods are performing on par with the proposed method in terms of the attack success rate and benign accuracy. The major difference seems to be the number of flipped bits. However, the paper in its current form does not state "why" the number of flipped bits seems to matter.
- The method is also performing on par with prior works in terms of the resiliency against defense mechanisms. Therefore, it is not really clear why this work is preferred over prior methods, i.e., does reducing the number of flipped bits matter in the attack?
- In addition to motivating the number of flipped bits, the authors need to also clarify why the current "single image" attack is preferred over other works where all images from a certain class are mapped to the attack target class. It seems like the proposed approach is in fact a special case of the latter scenario which is studied in prior works.
- The evaluated defense strategies are all passive, i.e., they are performed before the attack and therefore are not aware of the attack strategy. For a comprehensive examination, the authors should also compare with the defense in [1] which applied the defense at the inference phase, assuming a bit-flip attack may have occurred on the model parameters.

---

> ### Author Response · Authors · 2020-11-17
> **Response to Reviewer 4**
>
> Thank you for all your valuable comments and suggestions. We respond to your concerns below.
>
> **Q1&Q2**: Why the number of flipped bits seems to matter. It is not really clear why this work is preferred over prior methods, i.e., does reducing the number of flipped bits matter in the attack?
> \
> **A1&A2**: Thank you for pointing out this concern. Why it matters for the attack can be explained from three aspects as follows and we have added the explanation to the third paragraph in Section 3.2 in the revised paper.
> - The constraint on the number of flipped bits was originally introduced in Section 3.2 for ensuring the stealthiness. The curves in Appendix H verify that flipping too many bits hurts the PA-ACC.
> - Physical bit flipping techniques can be time consuming as discussed in [1,2]. Moreover, such techniques lead to abnormal behaviors in the attacked system ($e.g.$, suspicious cache activity of processes), which may be detected by some physical  detection-based defenses  [3]. Therefore, minimizing the number of flipped bits is essential to make the attack more efficient and practical. Besides, several recent works [2,4,5] also considered a minimum modification on the weights when performing the weight attack.
> - ASR, PA-ACC, and $\mathrm{N_{flip}}$ are three evaluation dimensions, which should be integrated to measure the attack performance, as commented by reviewer 3. Our method outperforms or at least is comparable with other methods in terms of ASR and PA-ACC and achieves less $\mathrm{N_{flip}}$, which demonstrates that the proposed method is overall a better attack.
>
> [1] Van Der Veen, Victor, et al. Drammer: Deterministic rowhammer attacks on mobile platforms. CCS, 2016.
> \
> [2] Zhao, Pu, et al. Fault Sneaking Attack: a Stealthy Framework for Misleading Deep Neural Networks. DAC, 2019.
> \
> [3] Gruss, Daniel, et al. Another Flip in the Wall of Rowhammer Defenses. IEEE S&P, 2018.
> \
> [4] Rakin, Adnan Siraj, Zhezhi He, and Deliang Fan. Bit-Flip Attack: Crushing Neural Network with Progressive Bit Search. ICCV, 2019.
> \
> [5] Rakin, Adnan Siraj, Zhezhi He, and Deliang Fan. TBT: Targeted Neural Network Attack with Bit Trojan. CVPR, 2020.
>
> **Q3**: Why the current "single image" attack is preferred over other works.
> \
> **A3**: Thank you for your insightful comment. We agree that "single image" attack is a special case of attacking more images to the target class. Nevertheless, studying this threat scenario is still very meaningful, as follows:
> - In some real attack scenarios, attacking a specific sample may meet the attacker’s requirement. For example, the attacker wants to manipulate the behavior of a face recognition engine on a specific input. The investigation of such a special case ("single image" attack) may yield a more stealthy attack method, therefore it is worthy of further exploration.
> - In addition, one of our contributions is to formulate the bit-flip based weight attack as a BIP problem, which can also be extended to attack more images. We will study it further in our future works.
>
> **Q4**: Evaluation with the defense which applied at the inference phase.
> \
> **A4**: Thanks for your valuable suggestion. The suggested defense at the inference phase is really insightful and deserves attention. In our paper, the experiment on the models with defense aims to verify the superiority of our optimization-based method in attacking more robust models. The robustness of model parameters is not well explored yet, especially for defense. If the reference [1] (it is missing in the current review) is pointed out, we would like to try to evaluate attacks under such defense.

---

### Official Review · AnonReviewer3 · 2020-10-29

**Rating:** 6
**Confidence:** 3

**Review:**

This paper proposes an ADMM based optimization method to conduct adversarial weight attack, and achieves superior or at least comparable performance compared with previous heuristic methods.

Pros:
1. Adversarial weight attack is an interesting research direction with important practical importance and deserve more studies.
2. The proposed method is mathematically sound. And it empirically outperforms or at least is comparable with previous state-of-the-art methods on undefended models, and consistently outperforms previous methods on defended models.

Cons:
I think this paper as an necessary step towards stronger adversarial weight attacks, which could be used as an evaluation method to benchmark future defense methods.

Some comments:
1. Table 1 and 2 may not be the best way to present the results. Considering there are three evaluation dimensions (PA-ACC, ASR and Nflip), I suggest the authors to add some pareto frontier figures. For example, fixing PA-ACC, plot the tradeoff curves between ASR and Nflip of different methods.
2. What are the time costs of different attacking methods?

---

> ### Author Response · Authors · 2020-11-17
> **Response to Reviewer 3**
>
> Thank you for your recognition of this work and all valuable comments and suggestions. We respond to your concerns below.
>
> **Q1**: I suggest the authors to add some pareto frontier figures.
> \
> **A1**: Thanks for your insightful suggestion. We have added a new section (Appendix H) to study the trade-off between the three metrics for our method in the revised paper.
> - For our proposed method, we can trade-off the three metrics by tuning the hyper-parameters. Therefore, following your suggestion, we study the trade-off between PA-ACC and $\mathrm{N_{flip}}$ and the trade-off between PA-ACC and ASR (the absence of the trade-off between $\mathrm{N_{flip}}$ and ASR is because the fixed PA-ACC is difficult to implement). A numerical example of the trade-off between PA-ACC and $\mathrm{N_{flip}}$ on CIFAR-10 is shown below, and all visualized curves and detailed descriptions have been added in Appendix H.
> | $\mathrm{N_{flip}}$ 	|  4.70 	|  5.33 	|  5.56 	|  9.94 	| 15.02 	| 29.90 	| 39.00 	| 43.07 	|
> |:-------------------:	|:-----:	|:-----:	|:-----:	|:-----:	|:-----:	|:-----:	|-------	|-------	|
> | PA-ACC              	| 85.49 	| 86.66 	| 88.07 	| 89.25 	| 89.58 	| 89.18 	| 88.87 	| 88.45 	|
> - Although the suggested way can present the results clearly, the trade-off curves of other methods could not be presented due to the below reasons. 1) T-BFA and GDA  greedily search the flipped bits until their attack goals are achieved, so there is no trade-off between PA-ACC, ASR, and $\mathrm{N_{flip}}$ for these two methods. 2) For TBT and FSA, they do not have the corresponding hyper-parameters to control the degree of meeting the three metrics, $e.g.$, ASR for TBT and $\mathrm{N_{flip}}$ for FSA.
>
> **Q2**: What are the time costs of different attacking methods?
> \
> **A2**: Thanks for your insightful suggestion. Following your suggestion, we have compared the results of the running time in Table 3 and added the discussion in Appendix C. Besides, the theoretical computational cost is $O\big(T_{outer} [ 2(N+1)CQ \cdot (C+T_{inner}) ] \big)$, which was originally analyzed in Appendix C.

---

### Official Review · AnonReviewer1 · 2020-10-29

**Rating:** 7
**Confidence:** 2

**Review:**

The paper describes a bit-flipping white-box attack on deployed neural network classifiers: given a model with quantized parameters, find a perturbation of the parameters bits such that the model with misclassify one specific example, while maintaining high accuracy on other examples.

The attack is formulated as a binary programmig problem where the parameter bits are the optimization variables and the objective function is an additive tradeoff between an effectiveness term (misclassification loss on the selected example) and a stealthness loss (classification loss on a batch of training examples), a constraint on the number of bit flips is also included. The optimization problem is solved by continuous relaxation using the Lp-box ADMM solver.

The paper reports experiments on various standard classifiers trained on CIFAR-10 or ImageNet, with different level of quantization. The proposed attack is compared to other weight attacks in the literature, and it achieves comparable or better attack success rate (a measure of effectiveness) and post-attack accuracy (a measure of stealthness).
There are also experiments on different values of hyperparameters and on more robust models (obtained either by a defense technique or by making the model bigger).

Overall I find this a valid contribution.

---

> ### Author Response · Authors · 2020-11-17
> **Response to Reviewer 1**
>
> Thanks for your recognition of this work and all valuable comments. Your comments are a great encouragement to us!

---

### Official Review · AnonReviewer2 · 2020-11-10
**Targeted Attack against Deep Neural Networks via Flipping Limited Weight Bits**

**Rating:** 6
**Confidence:** 4

**Review:**

The paper proposes an optimization-based algorithm for bit-flipping a limited number
of bits in a quantized / binarized deep-learning model, so that the prediction on a
target input example is flipped while the prediction on the other examples is as
untouched as possible. The problem is formulated as a binary integer programming (BIP)
problem, which is then solved using a recent ADMM-based technique. Experiments CIFAR-10
and ImageNet show that the proposed method outperforms the SOTA.

**Weak points:**
- The main shortcoming of this paper is the limitedness of the technical contributions.

- The hyper-parameter tuning is not clearly outlined / explained. This is problematic
  since there are quite a number of hyper-parameters. For example, I can count 4
  hyper-parameters in equation (9) alone (including ADMM stepsizes rho_i).

- It would be nice to have a back-of-envelop estimation of the complexity (running time,
  number of flops, etc.) of the proposed method, as a function of the maximum number of
  bits to flip (say).

**Small issues:**
- S. Boyd and co-workers have done a great job in popularizing ADMM. However, this method
  has been around at least since the 70s. Key papers to reference when talking about ADMM
  include:
  * Glowinski and Marroco (1975) "Sur l'approximation, par éléments finis d'ordre un, et
    la résolution par pénalisation-dualité d'une classe de problèmes de Dirichlet non
    linéaires dualite d'une classe de problemes de Dirichlet non linéaires"
  * Gabay and Mercier (1976) "A dual algorithm for the solution of nonlinear variational
    problems via finite element approximation"

**Strong points:**
- The strongest point in favor of this paper is that unlike the SOTA methods, the proposed
  method only flips to a very limited number of bits in the binarized DNN model, while
  achieving the same or higher accuracy.
- The experiments are very detailed and well-presented.

**Errors:**

- The equivalence in (6) doesn't seem to make sense. In the definition of $S_p$, $\hat{b}$ is an element of what ?

---

> ### Author Response · Authors · 2020-11-17
> **Response to Reviewer 2**
>
> Thank you for your recognition of this work and all valuable comments and suggestions. We respond to your concerns below.
>
> **Q1**: The hyper-parameter tuning is not clearly outlined / explained.
> \
> **A1**: We are sorry about this unclarity. Due to the space limit, the detailed hyper-parameter settings were originally provided in Appendix E.3. We re-clarify the hyper-parameter settings in our method here.
> - For the parameters in our objective function, $\lambda$ and $k$ affect the attack stealthiness and the attack success rate. We adopt a strategy for jointly searching $\lambda$ and $k$, which is specified in Appendix E.3. The slack variable $\delta$ can control the prediction probability on the attacked sample. The $\delta$ is fixed as 10 on CIFAR-10, and it is set to 3 and increased to 10 if the attack fails on ImageNet.
> - In our experiments, we increase ($\rho_1, \rho_2, \rho_3$) after each iteration to accelerate the practical convergence following the general setting. Other parameters of ADMM are fixed for all the experiments.
>
>
> **Q2**: Estimation of the complexity (running time, number of flops, etc.) of the proposed method, as a function of the maximum number of bits to flip.
> \
> **A2**: Thanks for your valuable suggestion.
> - The theoretical complexity analysis was originally provided in Appendix C. According to the overall computational cost $O\big(T_{outer} [ 2(N+1)CQ \cdot (C+T_{inner}) ] \big)$, there is no relationship between the computational complexity and the maximum number of bits to flip ($i.e.$, $k$). However, $k$ influences the constraint space, which may have a further impact on the running time. Therefore, to further study it, we evaluated the running time and the number of iterations on CIFAR-10 under different $k$, as shown in the following table. These results demonstrate that the running time and the number of iterations are not related to the parameter $k$ to some extent in our experiments.
> |   $k$   	|       5       	|       10       	|       15       	|       20       	|       25       	|       30       	|
> |:-------:	|:-------------:	|:--------------:	|:--------------:	|:--------------:	|:--------------:	|:--------------:	|
> | time(s) 	|  144.53±19.74 	|  147.24±14.09  	|  147.81±17.91  	|  150.57±14.43  	|  144.01±14.13  	|  139.55±30.64  	|
> | #iters  	| 1435.10±39.46 	| 1404.30±137.34 	| 1457.50±176.53 	| 1419.60±135.33 	| 1417.20±139.67 	| 1483.60±263.58 	|
> - Besides, we have reported the results of the running time of different methods in Table 3 and added the discussion in Appendix C for your reference.
>
>
> **Q3**: Small issues.
> \
> **A3**: We greatly appreciate you for pointing out the early key papers about ADMM, and we have added them to the third paragraph in Section 1 in our revised version.
>
>
> **Q4**: Errors.
> \
> **A4**: Sorry for the confusion and it has been modified to $\hat{b} \in (\mathcal{S}_b \cap \mathcal{S}_p)$ in the revised paper. Accordingly, $\hat{b}$ is the element of the intersection of the box constraint $\mathcal{S}_b$ and the $\ell_2$-sphere constraint $\mathcal{S}_p$. Thank you for pointing it out.

---

> > ### Comment · AnonReviewer2 · 2020-11-25
> > **Final comment**
> >
> > Thanks the the authors for responding to my answers. I'm still worried about an algorithm which has 4 or more hyper-parameters (with no clear way to figure them out). I just don't see how such an algorithm could have a reasonable out-of-the-box performance in the wild. The authors' response to this point that I raised in my review is unsatisfactory.
> >
> > I'm keeping my previous score of 6.

---

> > > ### Author Response · Authors · 2020-11-25
> > > **Response to Reviewer 2**
> > >
> > > Thank you for your additional feedback. We understand your concern about the hyper-parameters in our method and this concern is insightful. The introduced hyper-parameters can be divided into two parts, including (1) hyper-parameters in the $\ell_p$-Box ADMM algorithm, and (2) attacker-specified hyper-parameters in our attack formulation. We would like to clarify that our method is very practical across different models and datasets by providing more details about how to easily select those hyper-parameters in practice, as follows:
> > > -  Tuning hyper-parameters in ADMM: In our method, inspired by the $\ell_p$-Box ADMM [1] algorithm, our method also replaces the binary constraint in our attack formulation by the intersection of two continuous constraints, leading to more hyper-parameters ($i.e$., $\rho_1, \rho_2, \rho_3$) in ADMM. However, in Section 3.2 of [1], there are very detailed studies about the sensitivity to the $\rho$ parameters for convergence, and give a very specific and practical suggestion about how to tune the $\rho$ parameters to get a good converged solution. Our experimental settings about the $\rho$ parameters (see Appendix E.3 in our manuscript) exactly follow this suggestion. Here we simply repeat this suggestion, which consists of two parts:
> > > \
> > > __(1)__ Initializing $\rho$ from a small value $\rho_0$, then gradually increasing $\rho$ ($e.g$., $\rho \leftarrow 1.01 * \rho$), until a user-defined upper bound $\rho_{upper}$. The small initial value encourages more sufficient searches in the constraint space, while the gradual increase and the upper bound  ensure the final convergence.
> > > \
> > > __(2)__ The tuning of the extra hyper-parameters $\rho_0$ and $\rho_{upper}$ can be easily adjusted according to the value of the corresponding constraint violation (each $\rho$ corresponds to one constraint violation). If the violation value decreases very slowly along the iteration, then we can increase $\rho_0$ to accelerate the convergence; if it decreases very quickly, then we can decrease $\rho_0$ or $\rho_{upper}$ to encourage more sufficient searches.
> > > \
> > > In [1], $\ell_p$-Box ADMM has verified its effectiveness in image segmentation, graph matching and clustering. Moreover,  $\ell_p$-Box  ADMM is utilized to solve many diverse tasks, including model compression [2], MAP inference [3], hash code learning [4], etc. In our experiments, we also follow the suggestion in [1], and get very satisfied results. And the specific values of these hyper-parameters used in our experiments (see Appendix E.3), as well as the codes, have been provided to ensure the reproduction of the reported results. All these previous works and our experiments have verified the practical effectiveness of $\ell_p$-Box ADMM in many diverse tasks. Thus, we believe that there is no need to worry too much about the practical effectiveness of our method for different datasets and different DNN models.
> > > - Tuning attacker-specified hyper-parameters in our attack formulation: These parameters in the formulation correspond to the attacker’s preference as follows.
> > > \
> > > __(1)__ $\lambda$ controls the trade-off between attack effectiveness and stealthiness;
> > > \
> > > __(2)__ $k$ limits the maximum number of bit-flips;
> > > \
> > > __(3)__ $\delta$ manipulates the prediction probability of the attacked sample.
> > > \
> > > Utilizing the ADMM algorithm as demonstrated above, our attack can identify the flipped bits to satisfy the attacker’s preference by setting different attacker-specified hyper-parameters. The experiment results reported in the tables and ablation studies in Section 4.4 partially verify this point.
> > >
> > > In summary, the proposed method is very practical and can be applied to attack different models on different datasets. Thank you again for raising this point! We hope that the above explanations could somewhat alleviate your concern about the hyper-parameters.
> > >
> > > [1] Wu B, Ghanem B. $\ell_p$-Box ADMM: A Versatile Framework for Integer Programming[J]. IEEE transactions on pattern analysis and machine intelligence, 2018, 41(7): 1695-1708.
> > > \
> > > [2] Li T, Wu B, Yang Y, et al. Compressing convolutional neural networks via factorized convolutional filters[C]. CVPR, 2019.
> > > \
> > > [3] Wu B, Shen L, Zhang T, et al. MAP Inference Via $\ell _2 $-Sphere Linear Program Reformulation[J]. International Journal of Computer Vision, 2020: 1-24.
> > > \
> > > [4] Shen F, Xu Y, Liu L, et al. Unsupervised deep hashing with similarity-adaptive and discrete optimization[J]. IEEE transactions on pattern analysis and machine intelligence, 2018, 40(12): 3034-3044.

---

### Author Response · Authors · 2020-11-17
**Summary of updates in the revised paper**

We thank all the reviewers for the valuable comments and helping us to improve the paper. We have updated our paper according to reviewers' suggestions and summarize the revisions as follows:
- Added additional papers about ADMM to Section 1 as suggested.
- Changed equation (6) from $\hat{b} \in \mathcal{S}_b \cap \mathcal{S}_p$ to $\hat{b} \in (\mathcal{S}_b \cap \mathcal{S}_p)$.
- Added the explanation in Section 3.2 to clarify why reducing the number of flipped bits matters.
- Added the results of the running time of different methods in Appendix C.
- Added a new section (Appendix H) to study the trade-off between the three metrics for our method.

---

### Decision · Program_Chairs · 2021-01-07
**Final Decision**

**Decision:**

Accept (Poster)

**Comment:**

The major concerns about this paper are that (1) There are too many hyper-parameters, such as those needed for ADMM. I'd point out that there are adaptive variants of ADMM and heuristics methods for choosing optimization hyper-parameters, although it would be nice if the authors addressed these issues in the paper.  (2) Some reviewers are concerned that, compared to other related attacks, it’s unclear why flipping fewer bits is an important objective - an attacker might only care about poisoning performance and clean data performance.  The authors respond that flipping fewer bits makes the attack more effective when bits are manipulated by a physical method such as manipulating memory.  Despite these criticisms, reviewers agree that the paper is a well thought-out approach that improves the state of the art by some metrics.